# Synaptic interactions between stellate cells and parvalbumin interneurons in layer 2 of the medial entorhinal cortex are organized at the scale of grid cell clusters

Li-Wen Huang[1], Derek LF Garden[1,2,3], Christina McClure[1], Matthew F Nolan[1,2,4]*

[1]Centre for Discovery Brain Sciences, University of Edinburgh, Edinburgh, United Kingdom; [2]Simons Initiative for the Developing Brain, University of Edinburgh, Edinburgh, United Kingdom; [3]Institute of Medical Sciences, University of Aberdeen, Aberdeen, United Kingdom; [4]Centre for Statistics, University of Edinburgh, Edinburgh, United Kingdom

*For correspondence: mattnolan@ed.ac.uk

Competing interest: The authors declare that no competing interests exist.

## eLife Assessment

In this **compelling** study, the authors examine the interactions between stellate cells and PV+ interneurons in the medial entorhinal cortex. Huang et al. focus on the spatial distribution of synaptic inputs and demonstrate that closely located neuron pairs receive common inputs, suggesting a structured functional organization in the entorhinal cortex. Advanced dual whole-cell patch recordings further reveal patterns of postsynaptic activation, indicating intensive interactions within clusters of these neurons, with weaker interactions between clusters. These findings offer significant insights into the functional dynamics of the entorhinal cortex and the circuit mechanisms that shape grid cell activity. This study is **important** not only for the field of MEC and grid cells, but also for broader fields of continuous attractor networks and neural circuits.

**Abstract** Interactions between excitatory and inhibitory neurons are critical to computations in cortical circuits but their organization is difficult to assess with standard electrophysiological approaches. Within the medial entorhinal cortex, representation of location by grid and other spatial cells involves circuits in layer 2 in which excitatory stellate cells interact with each other via inhibitory parvalbumin expressing interneurons. Whether this connectivity is structured to support local circuit computations is unclear. Here, we introduce strategies to address the functional organization of excitatory-inhibitory interactions using crossed Cre- and Flp-driver mouse lines to direct targeted presynaptic optogenetic activation and postsynaptic cell identification. We then use simultaneous patch-clamp recordings from postsynaptic neurons to assess their shared input from optically activated presynaptic populations. We find that extensive axonal projections support spatially organized connectivity between stellate cells and parvalbumin interneurons, such that direct connections are often, but not always, shared by nearby neurons, whereas multisynaptic interactions coordinate inputs to neurons with greater spatial separation. We suggest that direct excitatory-inhibitory synaptic interactions may operate at the scale of grid cell clusters, with local modules defined by excitatory-inhibitory connectivity, while indirect interactions may coordinate activity at the scale of grid cell modules.

## Introduction

Computation in cortical circuits is shaped by synaptic interactions between excitatory principal cells and local interneurons (*Harris and Shepherd, 2015*; *Miller, 2016*; *Sadeh and Clopath, 2021*). Critical determinants of these interactions include the spatial extent over which given presynaptic neurons influence their targets, and whether connectivity is specific to subsets of neurons within the potential target population. These circuit properties can be challenging to assess experimentally. For example, while simultaneous patch-clamp recordings from pairs of neurons provide a gold standard for assessment of functional properties of synaptic connections (*Markram et al., 1997*; *Sjöström et al., 2001*), even with impressive advances in maximizing the numbers of simultaneously recorded neurons (*Peng et al., 2019*; *Wang et al., 2015*), their low throughput makes them less well suited to investigation of the spatial extent and organization of connectivity.

The medial entorhinal cortex (MEC) is a promising higher order cortical area for elucidation of principles for circuit level computations (*Moser and Moser, 2013*). The striking spatially periodic firing properties of grid cells found within the MEC has an anatomical organization that shares features with the columnar organization of sensory cortices (*Burgalossi and Brecht, 2014*; *Naumann et al., 2018*). Nearby grid cells have a similar spatial scale and orientation, which is maintained across superficial and deep cell layers (*Sargolini et al., 2006*). For grid cells at different dorsoventral locations the scale and orientation of their firing fields varies, with larger scales found in more ventral neurons (*Hafting et al., 2005*). This variability arises from discrete groups of grid cells that have firing fields with similar scale and orientation and form functional modules (*Barry et al., 2007*; *Gu et al., 2018*; *Stensola et al., 2012*; *Yoon et al., 2013*). Individual modules can be partitioned into cell clusters, with each cluster containing topographically ordered grid cells encoding the full range of possible spatial phases (*Gu et al., 2018*; *Heys et al., 2014*). A critical goal in determining the mechanisms that subserve grid firing is therefore to establish principles for organization of synaptic interactions at these anatomical scales.

Within layer 2 of the MEC, synaptic interactions between excitatory stellate cells (SCs) and inhibitory parvalbumin positive interneurons (PV+ INs) have received attention as a possible substrate for emergence of grid firing fields (*Beed et al., 2013*; *Buetfering et al., 2014*; *Couey et al., 2013*; *Miao et al., 2017*; *Pastoll et al., 2013*; *Shipston-Sharman et al., 2016*; *Widloski and Fiete, 2014*). Layer 2 of the MEC contains a high density of grid cells, with both stellate and pyramidal cells (PCs) in this layer having grid fields (*Domnisoru et al., 2013*; *Gu et al., 2018*; *Heys et al., 2014*; *Rowland et al., 2018*; *Schmidt-Hieber and Häusser, 2013*). Interactions between SCs appear to be primarily via PV+ INs (*Couey et al., 2013*; *Fuchs et al., 2016*; *Pastoll et al., 2013*; *Winterer et al., 2017*), and models based on this connectivity are able to account for grid firing and local theta-nested gamma frequency network activity (*Pastoll et al., 2013*; *Shipston-Sharman et al., 2016*; *Solanka et al., 2015*; *Widloski and Fiete, 2014*), while silencing PV+ INs reduces the spatial periodicity of grid firing in layer 2 (*Miao et al., 2017*). However, previous investigation of interactions between SCs and PV+ INs have focused on nearby pairs of cells, typically separated by <150 µm (*Couey et al., 2013*; *Dhillon and Jones, 2000*; *Fernandez et al., 2022*; *Fuchs et al., 2016*; *Grosser et al., 2021*; *Pastoll et al., 2013*; *Winterer et al., 2017*). We know much less about their functional synaptic interactions at distances at the scale of grid cell clusters or modules. Anatomical approaches, for example single neuron axo-dendritic reconstructions (*Burgalossi et al., 2011*; *Fuchs et al., 2016*; *Martínez et al., 2017*; *Quilichini et al., 2010*; *Tamamaki and Nojyo, 1993*), labeling of axon terminals of molecularly targeted SCs (*Sürmeli et al., 2015*), or high-density electron microscopy reconstructions (*Schmidt et al., 2017*), can constrain potential connectivity, but are limited here because they do not reveal functional interactions.

We set out to evaluate the extent and organization of functional synaptic connectivity between SCs and PV+ INs within layer 2 of the MEC across the scale of grid clusters and modules. To do so, we developed strategies to assess the functional organization of synaptic input to pairs of identified postsynaptic neurons from genetically defined populations of presynaptic neurons. Our results indicate that connectivity between SCs and PV+ INs is organized at the scale of grid cell clusters, such that nearby PV+ INs (or SCs) receive a large proportion, although not all, of their input from the same SCs (or PV+ INs), with this proportion dropping steeply with increasing cell separation. We also find evidence that indirect coordination can operate at longer distances, that networks of interacting SCs and PV+ INs have a modular organization, and that distinct PV+ INs coordinate SCs and nearby

pyramidal cells. Our results suggest highly structured and localized synaptic interactions within layer 2 of the MEC as candidates for the organization of spatial representations.

## Results

### Entorhinal stellate cells have extensive intralaminar axon collaterals

To investigate axonal substrates for intra-layer connectivity of SCs, we first reconstructed neurons from adult mice using methods that maximize recovery of axonal morphology following whole-cell recordings in brain slices (*Jiang et al., 2015*). We examined projections along the dorsoventral axis, recovered from neurons recorded in sagittal slices, and along the mediolateral axis, recovered from neurons recorded in horizontal slices (*Figure 1*).

Consistent with observations from labeling of presynaptic terminals (*Sürmeli et al., 2015*), we found axon collaterals of SCs in deep L1, L2, L3 close to the border with L4, and in L5b (*Figure 1A–E*). The dendrites of SCs were exclusively in L1-2 and the superficial part of L3. Individual SCs varied in whether they had axon collaterals in a particular layer (*Figure 1E*). While the total length of axon collaterals in superficial layers (L1-3) was greater than in deep layers (L5-6), there was no apparent relationship between the total length of superficial and deep axon collaterals (*Figure 1F*). For SCs with axon collaterals in L1-3, the average extent of their collaterals was >500 μm along the dorsoventral axis in sagittal slices, and >400 μm along the mediolateral axis in horizontal slices (*Figure 1G*).

These results suggest extensive axonal substrates for SCs to coordinate other neurons located along both mediolateral and dorsoventral axes in L1-2 of MEC. We next sought to establish whether this extensive axonal arborization supports functional organization of inputs to targets within L2.

### Asynchronous widefield activation of SCs reveals coordinated input to pairs of PV+ interneurons

To investigate potential functional organization of inputs to PV+ INs from SCs, we developed strategies for selective optogenetic activation of SCs and targeted patch-clamp recording from multiple PV+ INs (*Figure 2A*). As a first approach, we used ramp-like increases in the intensity of wide field illumination to drive asynchronous action potential firing by SCs expressing channelrhodopsin-2 (ChR2) (*Figure 2B*). As the timing of action potentials fired by SCs activated in this way differs across trials and between activated cells (*Figure 2B* and *Figure 2—figure supplement 1*), we reasoned that temporally correlated synaptic inputs to postsynaptic PV+ INs would indicate coordinated input from the activated SCs.

To validate this approach we simulated responses of pairs of cells to spike trains generated by action potentials recorded during ramp-like optogenetic activation of SCs (*Figure 2C* and *Figure 2—figure supplement 2*; see Methods). We compared models in which each postsynaptic neuron summed inputs from multiple presynaptic neurons but in which the number of common inputs varied (*Figure 2D–E*). For pairs of simulated PV+ INs receiving a greater fraction of common inputs, the average peak of the cross-correlation of their responses (CC_peak_mean) was higher and the variance in the lag of the cross-correlation (CC_lag_var) was lower (*Figure 2F* and *Figure 2—figure supplement 2*). Thus, cross-correlation of postsynaptic currents can index common synaptic inputs from upstream neurons activated by ramp-like stimuli.

We next recorded from pairs of PV+ INs, with cell bodies separated by distances between 63 μm and 1013 μm. Activation of SCs with light ramps induced excitatory postsynaptic currents (EPSCs) in all PV+ INs (*Figure 3A–B*). We observed substantial summation of light evoked EPSCs and individual events usually overlapped, suggesting inputs from many SCs converge onto each PV+ IN. Consistent with our model predictions (*Figure 2F* and *Figure 2—figure supplement 2F*), relatively high peak values of CC_peak_mean were associated with low values of CC_lag_var, indicating pairs that receive a common input (*Figure 3A and C*, *Figure 3—figure supplement 1*). In contrast, lower values of CC_peak_mean were associated with higher values of CC_lag_var, indicating pairs that do not share a substantial common input (*Figure 3B and D*, *Figure 3—figure supplement 1*). To establish whether the observed values of CC_peak_mean and CC_lag_var could have occurred by chance, we used distributions of values generated from shuffled datasets of 1000 pairs of cells that were recorded at different times. We classified pairs of PV+ INs as receiving coordinated synaptic input from SCs if their CC_peak_mean and CC_lag_var were above the 95th percentile of the shuffled data

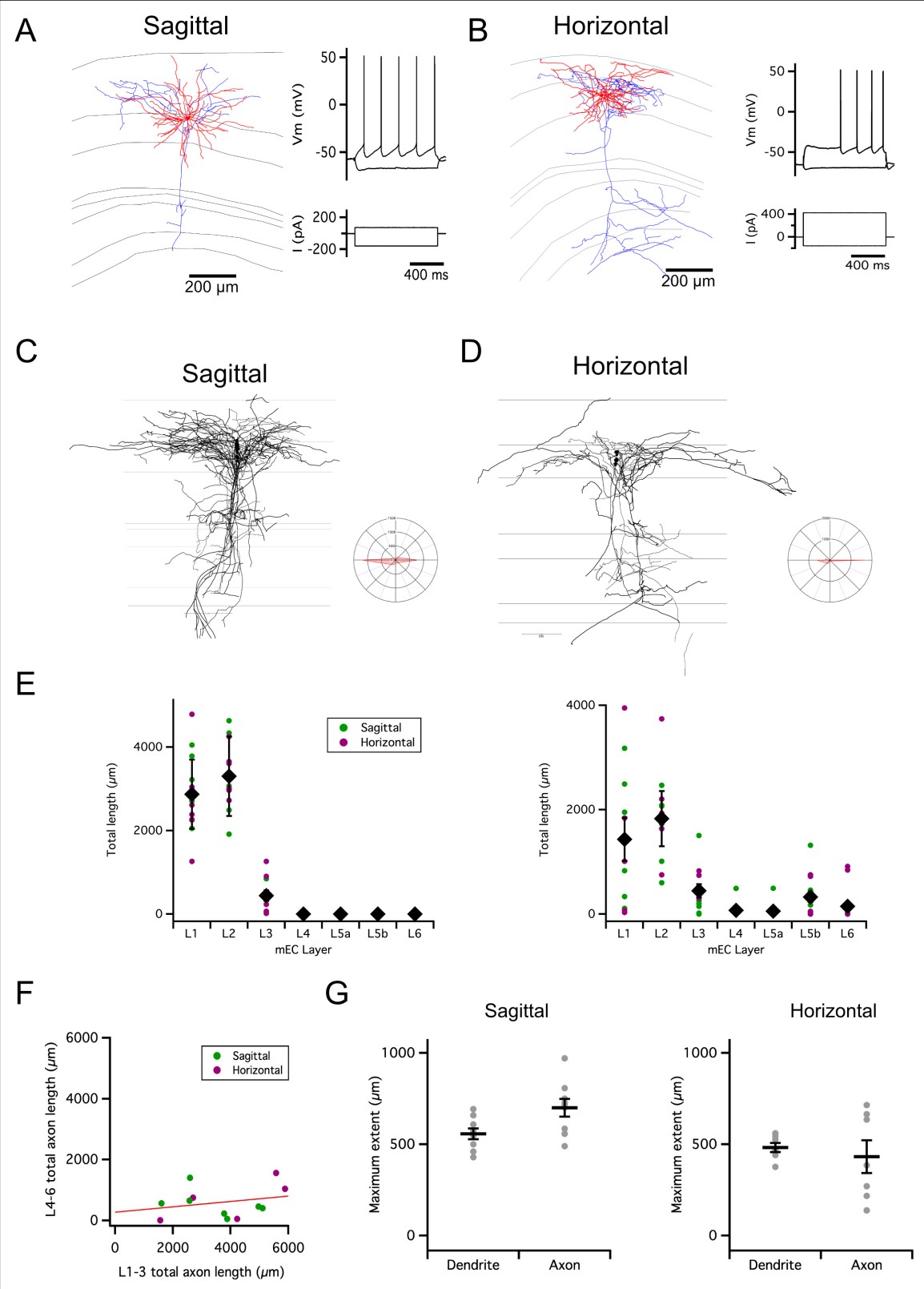

**Figure 1.** Organization of stellate cell axon collaterals. (**A–B**) Examples of SCs reconstructed following patch-clamp recording from a sagittal (**A**) and horizontal (**B**) brain slice. Axons are in blue and dendrites in red. Traces to the right indicate membrane potential responses (upper) to current steps (lower) used to establish the electrophysiological profile of the recorded neurons. (**C–D**). Overlay of all SC axons reconstructed from sagittal (n=8) (**C**) and horizontal (n=5) (**D**) slices with soma position registered relative to the L1/2 border (left). Angle plots for all SC axons (see wedge plots in Figure

*Figure 1 continued on next page*

*Figure 1 continued*

5 of *Sürmeli et al., 2015*) (right) indicate the dominance of axons that extend within layer 2. (**E**) Total length within each layer of dendrites (left) and axon collaterals (excluding the primary axon) (right). Colours distinguish data points from sagittal and horizontal slices, diamond symbols indicate the mean, and vertical bars indicate the standard error. The distribution of axons and dendrites depended on layer (p=1.72 x 10⁻¹¹, $F$=12.8, df = 6), but did not differ between axons and dendrites (compartment: p=0.33, $F$=0.94, df = 1; layer x compartment: p=0.35, $F$=1.13, df = 6) and was independent of the slice orientation (plane: p=0.71, $F$=0.14, df = 1; plane x layer: p=0.62, $F$=0.74, df = 6; plane x layer x compartment: p=0.29, $F$=1.24, df = 6; two-way repeated measures ANOVA). (**F**) Total axon length in superficial (L1-3) layers did not predict length in deep (L4-6) layers (linear model: p=0.30, adjusted $r^2$=0.014, $F$=1.17, df = 1, 11). (**G**) Maximum extent of L1-3 axons was greater than dendrites along the dorsoventral axis (sagittal sections; paired t-test: p=0.03, t=–2.7, df = 8), but similar along the mediolateral axis (horizontal sections; paired t-test: p=0.59, t=0.57, df = 6). Horizontal bars indicate the mean and vertical bars indicate the standard error.

set (corresponding to CC_peak_mean >0.27 and CC_lag_var <0.22, see *Figure 3—figure supplement 1C*). In this way, we identified 35/71 recorded pairs as receiving coordinated synaptic input from SCs (yellow points in *Figure 3E–F* and *Figure 3—figure supplement 1D*), but 0/1000 shuffled pairs (*Figure 3—figure supplement 1C*). This classification was consistent with grouping the cell pairs using a k-means classifier, which identified 29/71 coordinated pairs in one group and 36/71 uncoordinated pairs in a second group (*Figure 3—figure supplement 1E*). Together, these analyses show how asynchronous wide-field activation of SCs can be used to identify pairs of PV+ INs that share common synaptic input.

## Local direct and distant indirect coordination of PV+ interneurons by SCs

We found that coordination by SCs depends on the extent and direction of the separation between PV+ INs. Pairs of PV+ INs that received coordinated synaptic input from SCs were separated by distances up to 500 μm along the dorsoventral axis and up to 900 μm along the mediolateral axis (*Figure 3E–F*, *Figure 3—figure supplement 1A-B*). Along both axes, CC_peak_mean was reduced with cell separation indicating that coordination of nearby PV+ INs was greater and that coordination dropped more steeply with increasing separation. Pairs of PV+ INs for which we could not detect coordinated input were found along both axes across the full range of separations tested.

Because direct chemical and electrical synaptic connections between PV+ INs could in principle contribute to coordination of activity, we compared the extent of electrical interactions with coordination from SCs. We found that 3 of 49 pairs of tested PV+ INs were electrically coupled to one another (*Figure 3—figure supplement 2*). All three electrically coupled pairs were separated by distances <100 μm and received coordinated synaptic input during optogenetic activation of SCs. We did not find any evidence for inhibitory connectivity between 49 tested pairs of PV+ INs.

While the simplest explanation for shared synaptic input is that it arises directly from the activated ChR2 expressing neurons, it is also possible that coordination arises indirectly through multisynaptic pathways. Thus, excitatory outputs from SCs driven by ramping activation of ChR2 could depolarise other neuronal populations, which in turn either synapse with PV+ INs directly, or could synchronize the activity of SCs. Our cross correlation analyses of SC spiking indicate that synchronized firing of SCs is unlikely (*Figure 2—figure supplement 1*). Synchronization of SCs with pyramidal cells or other SCs also appears unlikely given that synapses from SCs to both populations are rare and when detected typically generate small postsynaptic responses (*Couey et al., 2013*; *Fuchs et al., 2016*; *Pastoll et al., 2013*; *Winterer et al., 2017*). However, it is possible that SCs drive activation of interneuron populations that in turn could mediate correlated inputs to the recorded PV+ INs.

To test this possibility we evaluated the effect of blocking fast inhibitory responses mediated by GABA$_A$ receptors using GABAzine. We found that for pairs of PV+ INs that received coordinated input in control conditions and were separated by <150 μm, when GABA$_A$ receptors were blocked their responses to ramp activation of SCs were similar (*Figure 4A–B and D–E*). By contrast, for PV+ INs separated by >150 μm, CC_peak_mean was typically reduced and CC_lag_var increased, and in many cases responses to ramp activation of SCs were no longer consistent with the PV+ INs receiving coordinated inputs (*Figure 4C–E*), indicating that in control conditions their shared input resulted from recruitment of inhibitory neurons.

Together with previous data indicating strong connections from SCs to PV+ INs and weak or absent connections to other SCs or to pyramidal cells (*Couey et al., 2013*; *Fernandez et al., 2022*; *Fuchs*

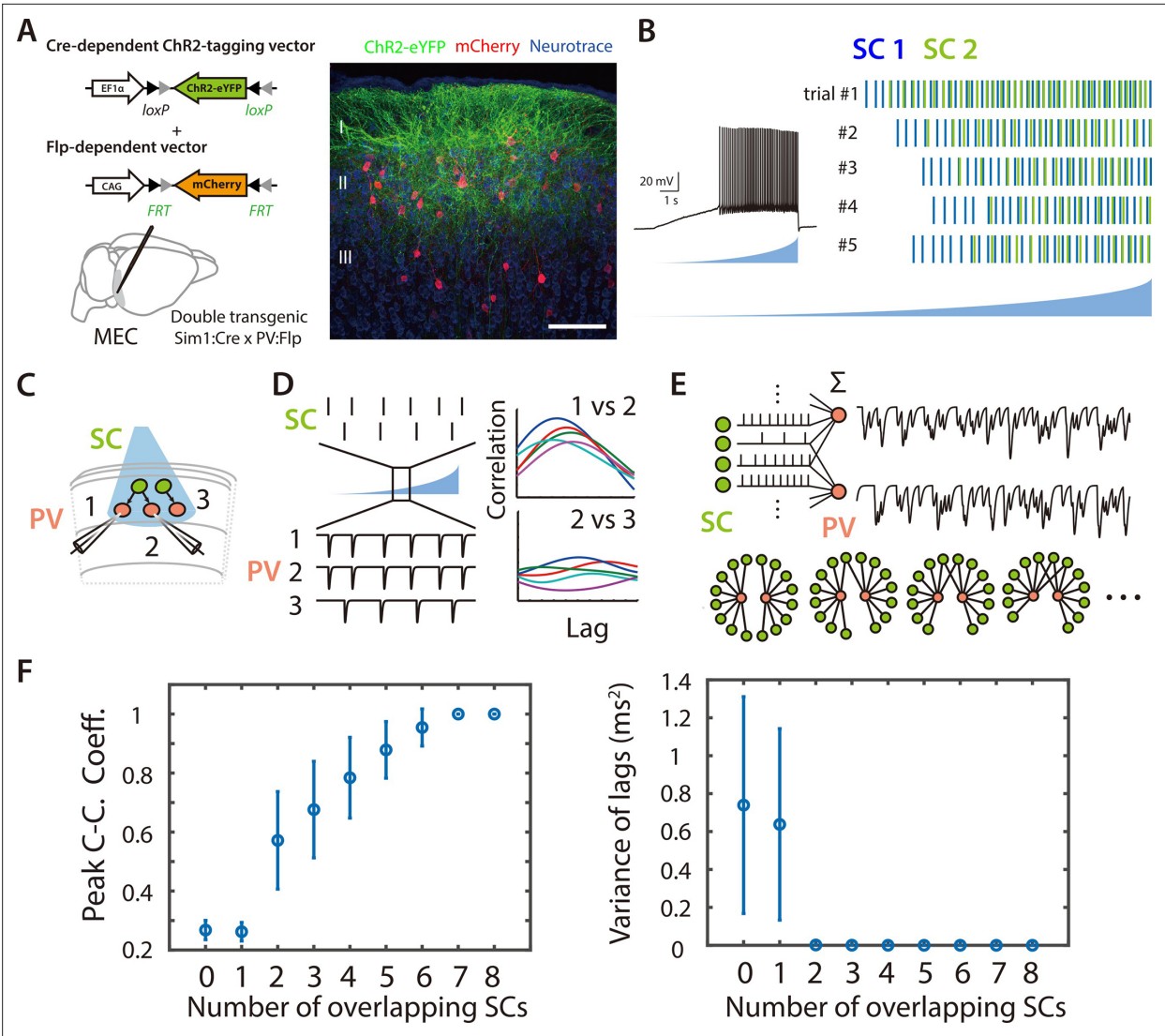

**Figure 2.** Asynchronous widefield activation strategy for investigating coordination between SCs and PV+ INs. (**A**) Experimental strategy (left). To enable optical activation of SCs and targeted recording from PV interneurons, AAVs that express ChR2-eYFP conditionally on the presence of Cre and mCherry conditionally on the presence of Flp were injected into Sim1:Cre x PV:Flp double transgenic mice. Confocal imaging of fixed brain sections (right) reveals labeled SCs (green) and PV+ interneurons (red). Scale bar = 100 μm. (**B**) Example of spiking (left) and spike rasters (right) from two simultaneously recorded SCs (right panel) in response to ramp-modulated optical activation (blue). Note that spike firing by the two SCs is asynchronous. (**C**) Schematic of the simulated recording configuration used for evaluation of input from SCs to PV+ interneurons. (**D**) Schematic of spike rasters during ramp-modulated optical activation of SCs (upper left) and corresponding synaptic currents from the 3 PV interneurons in (lower left) for the circuit configuration in (**C**). Cross-correlation of EPSCs (right) recorded from pairs of PV cells receiving inputs from the same neuron (1 vs 2) are expect to show large and invariant peaks, whereas for PV+ interneurons receiving input from different SCs (2 vs 3) their peaks should be small and have variable latencies. (**E**) Schematic of the procedure for convolving EPSC waveforms with experimentally recorded spike times of ChR2-expressing SCs responding to the ramp-modulated light stimulus (upper). The proportion of overlapping SCs in each simulation is varied while the total SC input to each PV+ interneuron remains constant (lower). (**F**) Peak cross-correlation coefficient (left panel) and variance of the cross-correlation lag (right panel) as a function of number of overlapping SCs in each simulation. Mean and standard deviation for each group were calculated from 100 simulations.

The online version of this article includes the following figure supplement(s) for figure 2:

**Figure supplement 1.** Optical ramp activation drives asynchronous activation of SCs.

**Figure supplement 2.** Simulation of SC inputs to PV+ interneurons.

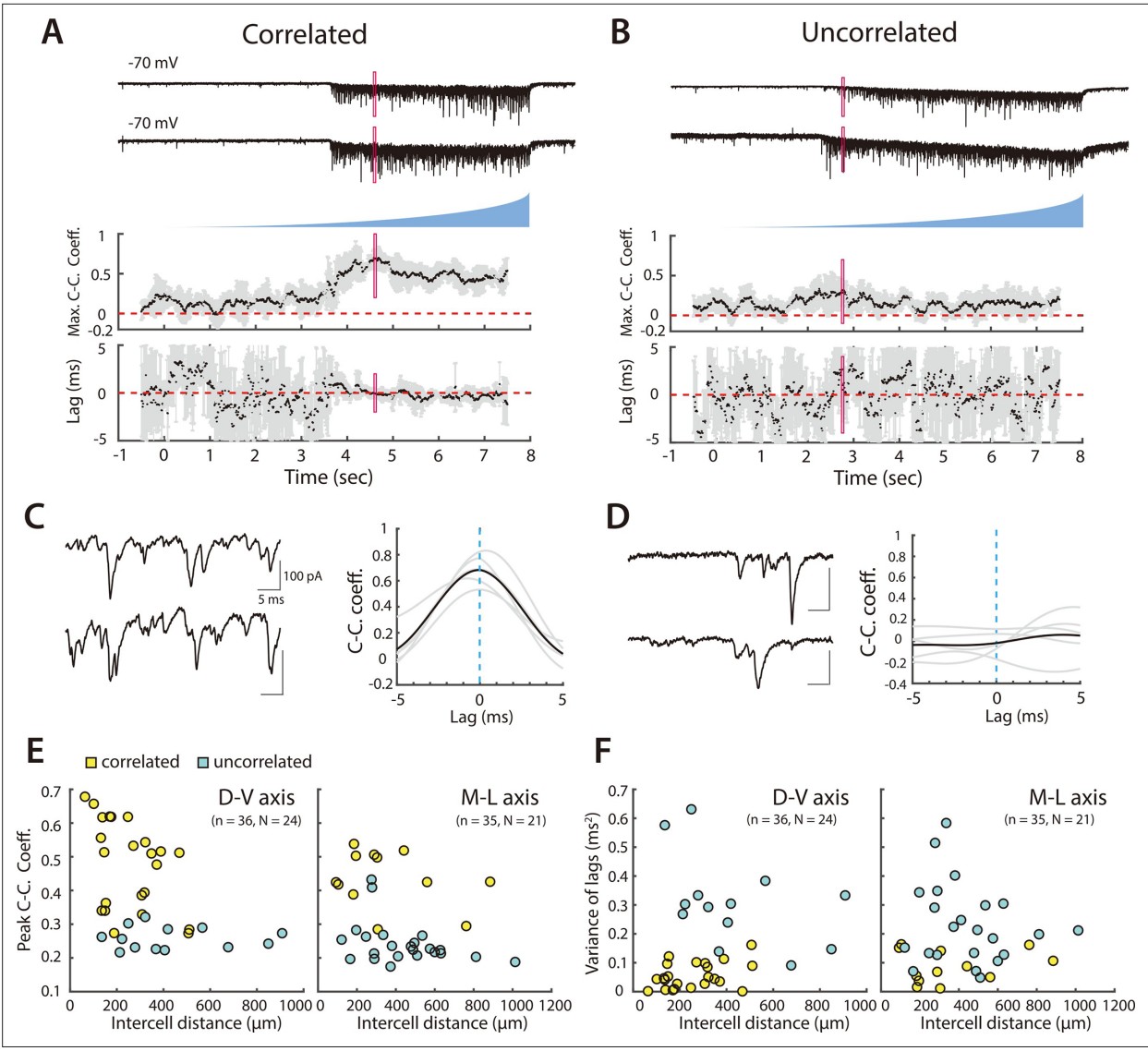

**Figure 3.** Organization of correlated input from SCs to PV+ interneurons. (**A, B**) Examples of simultaneously recorded EPSCs (upper) from pairs of PV+ interneurons and corresponding measures of the maximal correlation coefficient and correlation lag variance in response to asynchronous activation of SCs. In (**A**) the correlation coefficient increases and the lag variance decreases with optical activation of SCs, indicating that the interneuron pair receives input from a shared SC population, whereas in (**B**) both parameters are independent of SC activation, indicating that each interneuron receives input from distinct SC populations. Maximum cross-correlation coefficients and cross-correlation lag variance are plotted as means for each window (black dots) and corresponding standard deviations (grey bars). Red rectangles indicate regions shown on an expanded timebase in (**C, D**). (**C, D**) Current traces (left) for the analysis window in which the mean peak cross-correlation coefficient was greatest (indicated by the red rectangles in panels A and B), and corresponding mean (bold) and individual (grey) cross-correlograms from five consecutive trials (right). (**E, F**) Peak cross-correlation coefficient (**E**) and variance of the corresponding lag (**F**) as a function of cell body separation along the dorsoventral axis (left panel) and mediolateral axis (right panel). Pairs indicated by yellow points were identified as receiving correlated inputs. Peak cross-correlation coefficients were correlated with intercell distance along the dorsoventral (adjusted $R^2$=0.19, p=0.0046, $F_{1,34}$ = 9.2) and mediolateral axes (adjusted $R^2$=0.09, p=0.045, $F_{1,33}$ = 4.3), whereas variance of the lags did not show a detectable dependence on distance (horizontal: adjusted $R^2$=0.006, p=0.27, $F_{1,34}$ = 1.2; vertical: adjusted $R^2$=–0.03, p=1.0, $F_{1,33}$ = 0.0).

The online version of this article includes the following figure supplement(s) for figure 3:

**Figure supplement 1.** Distribution of PV+ INs receiving correlated input from SCs.

**Figure supplement 2.** Testing for electrical and chemical synaptic connections between PV+ INs.

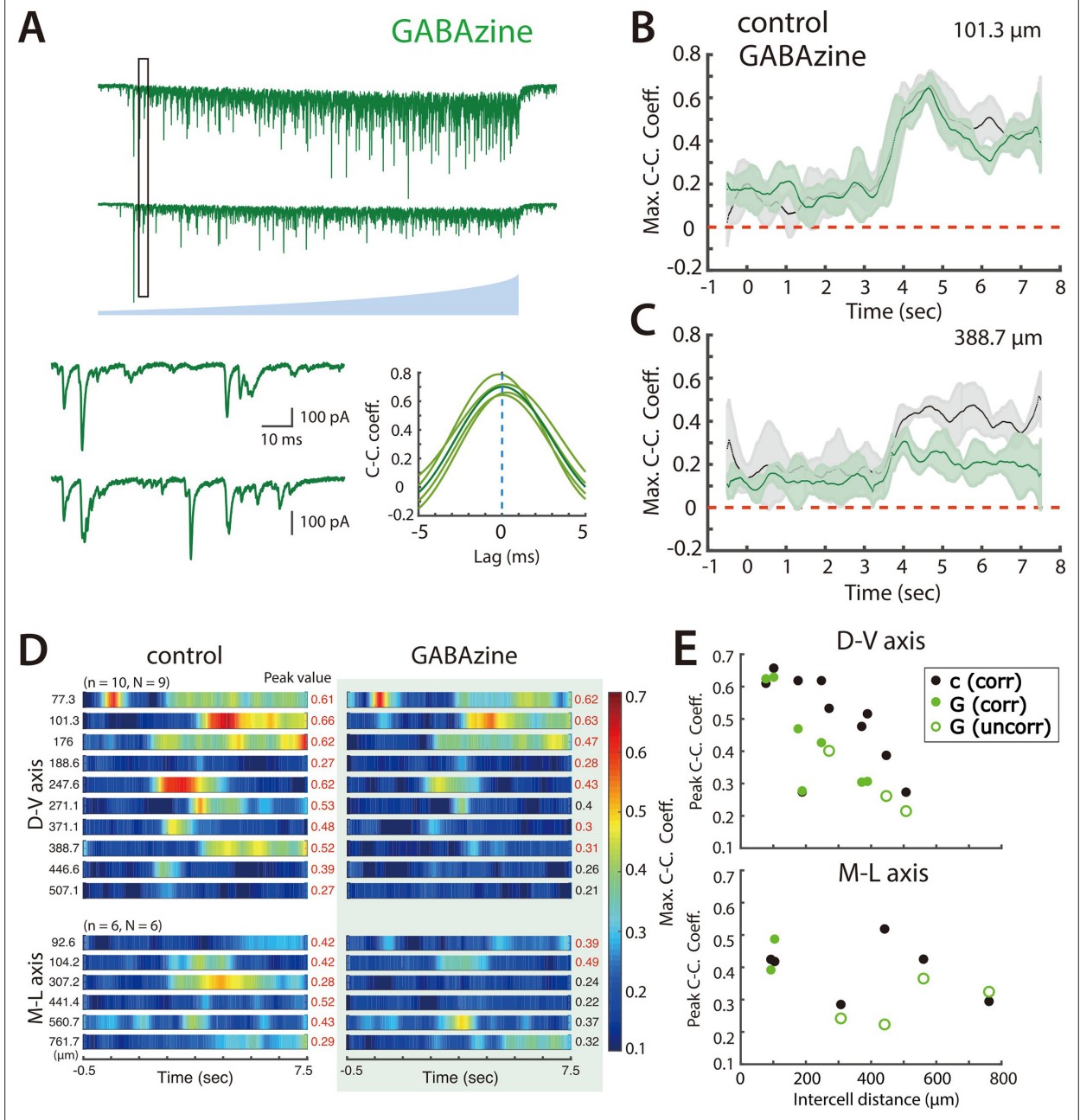

**Figure 4.** Synaptic inhibition differentially contributes to coordination by SCs of nearby and distant PV+ interneurons. (**A**) Simultaneously voltage-clamp recording, in the presence of GABAzine to block fast inhibition, of the responses of a pair of PV+ interneurons (upper traces) to ramping light (blue) activation of ChR2 expressing SCs. The somata of the recorded neurons were approximately 101 µm apart. The black rectangles indicate the time frame of the expanded traces and corresponding cross-correlograms (lower). (**B–C**) Maximum cross-correlation coefficients as a function of time since light onset in control (grey) and GABAzine (green) conditions for the cell pair shown in panel A (**B**) and for a cell pair with cell bodies separated by approximately 389 µm (**C**). (**D**) Heat maps of mean maximum cross-correlation coefficient as a function of time since light onset for the simultaneously recorded PV+ interneurons in the control condition (left panel) and in the presence of GABAzine (right panel). Distances separating pairs of recorded cells are indicated to the left and peak values of the maximum cross-correlation coefficient to the right. (**E**) The changes in peak cross-correlation coefficient between control (**c**) and GABAzine (**G**) conditions plotted as a function of distance between cells in each pair (corr = classified as correlated input, uncorr = classified as uncorrelated input).

*et al., 2016*; *Pastoll et al., 2013*; *Winterer et al., 2017*), these results suggest that coordination of PV+ INs involves multiple mechanisms with distinct anatomical organization. Direct electrical synaptic interactions are restricted to PV+ INs separated by <100 µm. In contrast, coordination of PV+ INs by SCs is more extensive and is consistent with previously described direct connections from SCs to PV+ INs (*Couey et al., 2013*; *Fuchs et al., 2016*; *Pastoll et al., 2013*; *Winterer et al., 2017*), and with additional indirect inhibitory pathways (*Fernandez et al., 2022*), which appear to increase the spatial extent of coordination.

## Focal activation of SCs generates common input to nearby PV+ interneurons

To corroborate our conclusions from responses to asynchronous widefield activation of SCs, we adopted a complementary approach in which we investigated responses of PV+ INs to activation of SCs using focal laser stimulation. We applied stimuli across a 7x10 array of locations in the superficial MEC, with each location separated by 20 µm (*Figure 5A*). We reasoned that if focal light pulses asynchronously activate restricted populations of ChR2 expressing SCs, then simultaneous postsynaptic responses in pairs of PV+ INs will identify inputs from the same SC.

We validated this approach by recording from SCs expressing ChR2 (*Figure 5—figure supplement 1A*). At threshold stimulation intensities the timing of action potentials was variable between trials (*Figure 5—figure supplement 1A–C*) and action potentials were typically triggered from a single location (1.63±0.74 locations, proportion of responses 0.53±0.36; n=8 cells, N=4 mice, *Figure 5—figure supplement 1D*). With increasing stimulation intensity, the latency to spike became shorter and the trial-to-trial variability in spike timing was reduced but not abolished (*Figure 5—figure supplement 1A–D*). The number of locations from which action potentials were initiated increased with increasing stimulation intensity, although at the maximal stimulation intensity action potentials were still typically triggered at relatively few locations close to the soma (4.5±2.78 locations). Therefore, because focal stimulation activates SCs with latencies that vary from trial to trial, and individual SCs vary in their light intensity thresholds, postsynaptic events occurring simultaneously in pairs of PV+ INs are likely to originate from the same presynaptic SC. Because activation of SCs was sparse and postsynaptic responses unlikely to be sufficient to drive spiking (cf. *Couey et al., 2013*; *Fuchs et al., 2016*; *Pastoll et al., 2013*; *Winterer et al., 2017*), it is unlikely that multisynaptic pathways would be activated through this focal stimulation approach.

To use focal stimulation to assess the extent to which pairs of PV+ INs received common input, we focused on response to stimulus intensities that evoked 335–908 EPSCs across all trials, which corresponds to approximately 7 x the threshold for a detectable response in either of the pair of PV+ INs. We calculated a correlation probability (Pc) (*Figure 5B–E*, see also *Yoshimura et al., 2005*), which we determined by finding the peak of the summed cross-correlations for all events across all stimulus locations, subtracting the corresponding peak from a shifted dataset, and normalizing by the estimated number of evoked events (see Methods). To identify pairs with Pc unlikely to be generated by chance, we generated a shuffled distribution by calculating Pc for synthetic pairs of PV+ INs recorded at different times. We identified pairs with Pc > the 95th percentile of the shuffled distribution as receiving a common input (*Figure 5—figure supplement 1F*).

When we investigated the variation in the correlation probability as a function of separation between PV+ INs, we found high correlation probabilities indicative of common synaptic input only for PV+ INs within 200 µm in sagittal slices and within 400 µm in horizontal slices (*Figure 5F*). In this experiment low values of $P_C$ could reflect local network structure, such that some nearby PV+ INs rarely receive common input from the same SCs, or it may reflect undersampling of the possible SC inputs to each pair of PV+ INs. To distinguish these possibilities, we compared correlation probabilities between stimulus intensities over which the number of postsynaptic events in PV+ INs differs across an approximately two-fold range. We found that the correlation probability was independent of the number of activated SCs (*Figure 5—figure supplement 1H*), suggesting that differential sampling of SC axons does not account for the variation in correlation probability between nearby pairs of PV+ INs.

These data show that focal laser stimulation of ChR2 expressing presynaptic neurons in combination with simultaneous recordings from postsynaptic neurons can assess the extent to which pairs of neurons receive common inputs. They support the conclusion from the previous widefield activation experiments that connections from SCs to PV+ INs are locally structured.

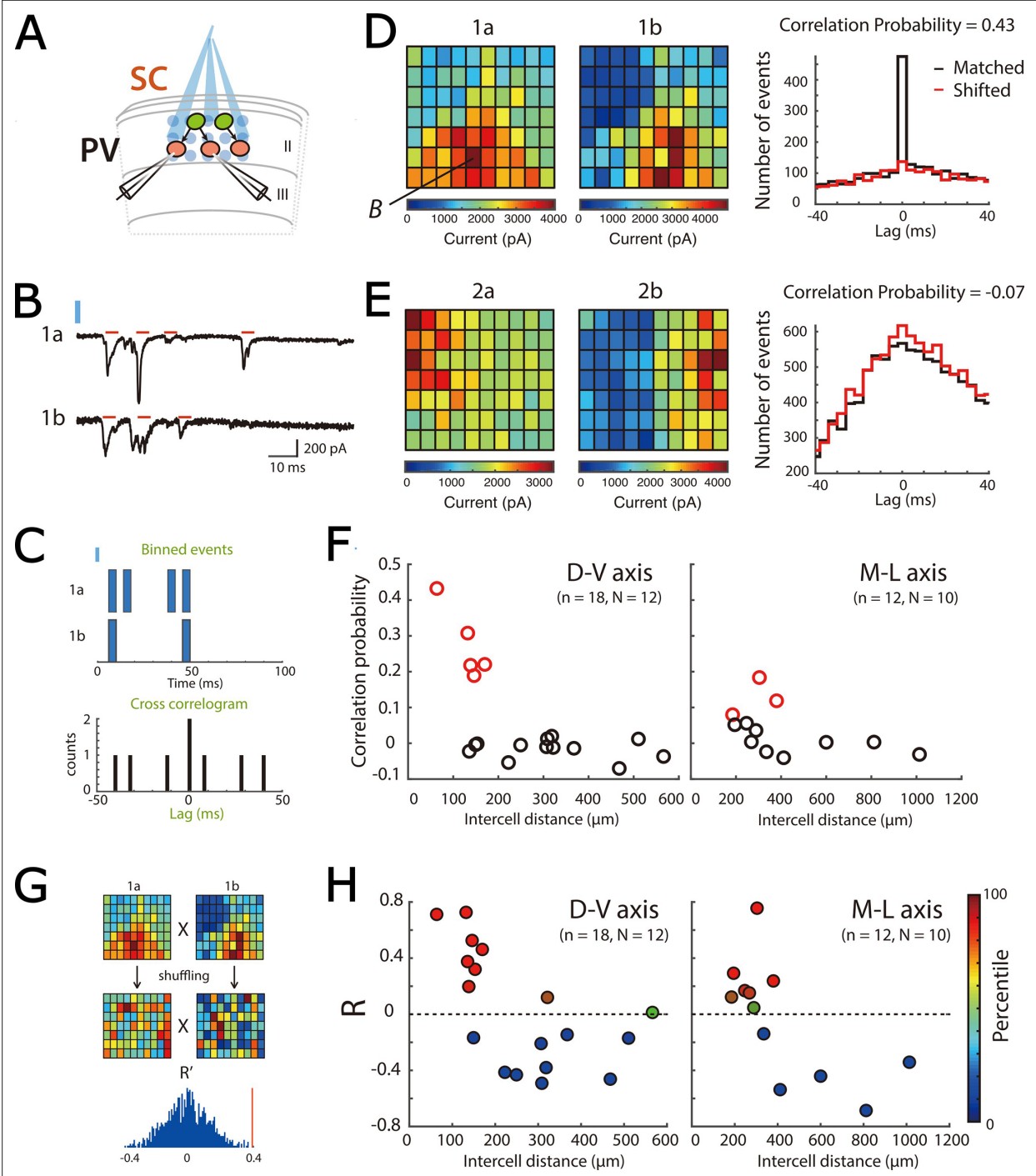

**Figure 5.** Focal activation of SCs confirms local direct connectivity to PV+ interneurons and reveals spatial organization. (**A**) Schematic of experimental set up. Simultaneous whole-cell voltage-clamp recordings were made from pairs of PV+ inerneurons. ChR2-expressing SCs were activated by focal laser stimulation. (**B**) Examples of light-evoked current responses from a pair of simultaneously recorded PV cells. Red bars indicate detected events, blue bar indicate time of laser activation. (**C**) Schematic of analysis of spatial correlation of active input locations. (**D–E**) Heat maps (left) for each of two simultaneously recorded neurons (**a and b**) show total current amplitude at each stimulation location (x and y axes identify locations within the stimulation grid, example traces from the location marked B are shown in panel B). Histograms (right) show correlograms for matched (black) and shifted (red) events. A high correlation probability (**D**) is indicative of neurons receiving common inputs, whereas a low correlation probability (**E**) is indicative of neurons receiving distinct inputs. (**F**) Correlation probability as a function of intercell distance along the dorsal-ventral axis (left panel; adjusted $R^2$=0.33, p=0.0072, $F_{1,16}$ = 9.5) and the medial-lateral axis (right panel; adjusted $R^2$=0.13, p=0.14, $F_{1,10}$ = 2.6). Cell pairs with correlations unlikely to have arisen by chance are indicated in red. (**G**) Schematic of the strategy for comparing the spatial organization of inputs from SCs to pairs of PV+ INs. Pearson's

*Figure 5 continued on next page*

*Figure 5 continued*

coefficient (**R**) was calculated following correlation of the heat maps for each cell, and for 100 pairs of shuffled heatmaps. In this example, the R from the original data (red vertical line) is above the 95th percentile of the shuffled data. (**H**) Pearson's coefficient (**R**) following correlation of spatial maps as a function of intercell distance along the dorsal-ventral axis (left panel; adjusted $R^2$=0.31, p=0.0092, $F_{1,16}$ = 8.8) and the medial-lateral axis (right panel; adjusted $R^2$=0.39, p=0.017, $F_{1,10}$ = 8.16). Data are color-coded according to their percentile in the shuffled data.

The online version of this article includes the following figure supplement(s) for figure 5:

**Figure supplement 1.** Focal activation of SCs.

## Organization of SCs projecting to PV+ interneurons is non-uniform and suggests discrete clustering

If grid cell clusters result from locally organized connectivity of neural circuits in the MEC, then we would expect to find a corresponding organization of functional SC to PV+ IN connections. In this case, stimulus locations that activate inputs to pairs of PV+ INs should either be similar, if the PV+ INs are both part of the same functional zone, or distinct if they are each members of different zones. Alternatively, if there is no clear spatial organization of SC to PV+ INs connections, then the correspondence between active stimulus locations for pairs of PV+ INs should have a random distribution.

To distinguish these possibilities, we built maps for responses of individual PV+ INs across all stimulation sites (*Figure 5D–E*). We then calculated scores for cross correlations (Pearson's coefficient (R)) between stimulus maps of simultaneously recorded PV+ INs and compared them to scores generated by shuffled maps for each pair (*Figure 5G*). We found that most cross-correlation scores differed significantly from chance, and maps could be positively or negatively correlated (*Figure 5H*). Examination of maps indicated that strong correlations (or anti-correlations) were associated with clear borders (*Figure 5D–E*). The correlation score was independent of the stimulus intensity, indicating that rather than reflecting a particular subset of neurons it was a consistent property of the network (*Figure 5—figure supplement 1I*). When we compared the spatial correlation scores with input correlation probabilities for the same pairs of PV+ INs, we found that pairs with strong spatial correlations typically had higher temporal input correlations, whereas pairs with low temporal input correlations also had low spatial input correlations (*Figure 5—figure supplement 1G*).

Together, these data suggest a structured functional organization of input from SCs to PV+ INs, provide evidence for anatomical boundaries between populations of SCs, and suggest that for pairs of PV+ INs that receive common input from SCs, common connectivity is not all to all, but arises from distinct subsets of SCs.

## PV+ interneurons have extensive axonal arborizations restricted to layer 2

How is inhibitory feedback in the reverse direction, from PV+ INs to SCs, organized? To assess the possible extent of influence of individual PV+ INs, we first investigated the axonal and dendritic organization of PV+ INs filled during targeted recordings from neurons in PV$^{Cre}$ (or PV$^{Flp}$) mice labeled by Cre- (or Flp-) dependent reporters. All labeled neurons recorded from were in layer 2 and had fast spiking action potential firing patterns that are characteristic of PV+ INs (*Figure 6A–B*). In contrast to the relatively sparse axon collaterals of SCs, almost all PV+ INs had dense axonal arborizations within layer 2 and dendrites primarily within layers 1 and 2 (*Figure 6A–F*). A single PV+ IN (n=1/24) had very different axonal morphology, with dendrites and axon that extended into layer 1. We did not observe axonal projections to deep layers of the MEC. We focus our anatomical analysis on the PV+ INs with axons restricted to layer 2. The maximal dorsoventral and mediolateral extent of the axons of these neurons was on average >600 µm, and in both orientations was greater than the corresponding extent of their dendrites (sagittal p=3.15 x $10^{-6}$ n=17, Horizontal p=8.9 x $10^{-4}$ n=7; *Figure 6G*).

These observations suggest that axonal output from most PV+ INs with cell bodies in L2 of the MEC is restricted within L2, which is in agreement with recent observations (*Fuchs et al., 2016*; *Grosser et al., 2021*). In addition, they show that axonal projections are extensive, often approaching 1000 µm.

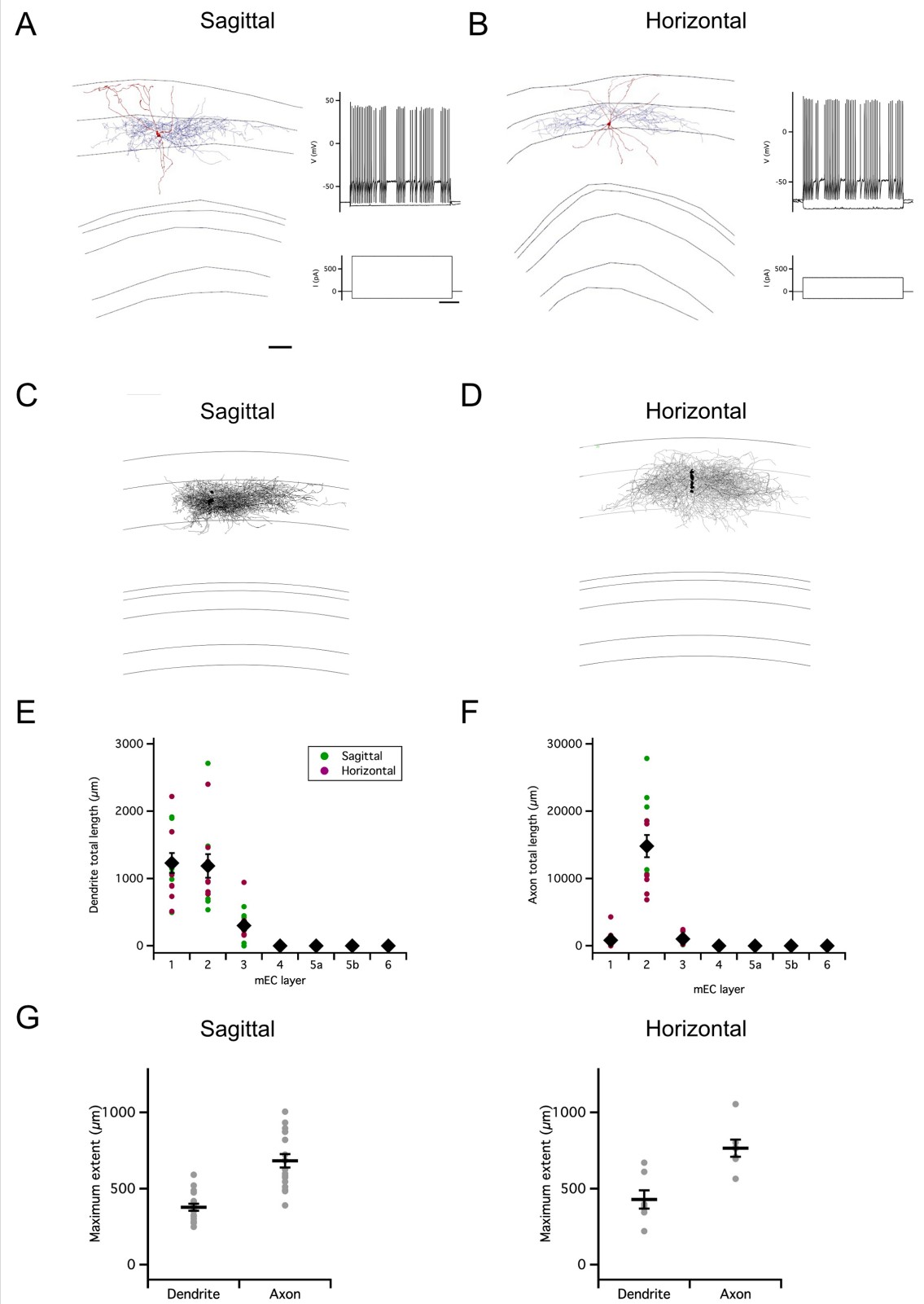

**Figure 6.** Organization of PV+ interneuron axons. (**A–B**) Examples of reconstructed axonal (blue) and dendritic (red) morphology of PV+ interneurons in sagittal (**A**) and horizontal slices (**B**) (scale bar 200 μm). Axons are in blue and dendrites in red. Traces to the right indicate membrane potential responses (upper) to current steps (lower) used to establish the electrophysiological profile of the recorded neurons (scale bar 200ms). (**C–D**) Overlay of all PV interneurons reconstructed from sagittal (**C**) and horizontal (**D**) slices with soma position registered relative to the L1/2 border (left). Each

*Figure 6 continued on next page*

*Figure 6 continued*

reconstruction was aligned to the border between L1 and L2. (**E–F**) Total length of PV+ IN dendrites (**E**) and axons (**F**) in each layer of the MEC. Data is from the reconstructions in (**C–D**). Colours distinguish data points from sagittal and horizontal slices, diamond symbols indicate the mean, and vertical bars indicate the standard error. The distribution of lengths depended on layer (layer: $p=<2 \times 10^{-16}$, $F=45$, df = 6), differed between axons and dendrites (compartment: $p=4.2 \times 10^{-8}$; $F=33.33$, df = 1; compartment x layer:$<2 \times 10^{-16}$, $F=37.21$, d=6) and between slice orientations (plane: $p=7.6 \times 10^{-5}$, $F=16.54$, df = 1; plane x layer: $p=2.7 \times 10^{-12}$, $F=13.54$, df = 6; plane * layer * compartment: $p=6.6 \times 10^{-12}$, $F=13.15$, df = 6; repeated measures ANOVA). (**G**) Maximum extent of dendrites and axons along the dorsoventral (sagittal, n=17) and mediolateral (horizontal, n=7) axes of the MEC. The maximal axonal extent was greater than that for dendrites in the sagittal and the horizontal planes (sagittal: $p=3.15 \times 10-6$, $t=-7.0$, df = 16; horizontal: $p=0.00089$, t - 6.1, df = 6; paired t-test). Horizontal bars indicate the mean and vertical bars indicate the standard error.

### Extent of coordination of SCs by PV+ interneurons

To address the extent to which the organization of inhibition from PV+ INs to SCs mirrors that of coordination in the opposite direction, we targeted expression of ChR2 to PV+ INs and recorded simultaneously from pairs of SCs (*Figure 7A–B* and *Figure 7—figure supplement 1A–B*). When populations of PV+ INs were asynchronously activated using ramp stimuli, recordings from SCs revealed inhibitory synaptic responses. In contrast to the responses of PV+ INs to activation of SCs, PV+ IN-driven synaptic currents recorded from SCs were relatively sparse and often of large amplitude (*Figure 7C*).

A subset of the pairs of SCs showed responses to activation of PV+ INs that were highly correlated with little variation in their lag, indicating that these pairs shared input from the same PV+ INs (*Figure 7C* and *Figure 7—figure supplement 1*). We considered all pairs with peak correlation >0.35 and corresponding lag variance <0.1 ms² to receive common input. Classifying cell pairs into two groups using a k-means algorithm gave a similar but slightly less conservative separation (*Figure 7—figure supplement 1D*). The cell bodies of pairs receiving common input were separated by <300 μm along dorsoventral and mediolateral axes (*Figure 7D* and *Figure 7—figure supplement 1E*). In contrast, pairs of neurons that did not demonstrate correlated input were found at all distances tested, indicating that close proximity is not sufficient for SCs to receive input from the same PV+ INs.

Pyramidal cells in layer 2 of the MEC have a topographical organization that is distinct from SCs and receive distinct inhibitory input from CCK positive basket cells (*Varga et al., 2010*). To test whether the same PV+ INs project to pyramidal cells and SCs, or if the two cell types are also distinguished by the origin of their input from PV+ INs, we evaluated responses of simultaneously recorded pairs of cells to activation of PV+ INs. We did not find any pairs of SCs and pyramidal cells with correlated responses that suggested a common input (*Figure 7E* and *Figure 7—figure supplement 1C and F*), indicating that they receive input from distinct populations of PV+ INs.

## Discussion

We have introduced new approaches for assessing the functional organization of genetically identified neuronal populations. Using these approaches, we find that direct functional interactions between SCs and PV+ INs in L2 of the MEC appear localized largely to within ~250 μm. The probability of shared connectivity in each direction is variable, with functional connections from SCs to PV+ INs appearing to be demarcated by distinct boundaries. We speculate that directly interacting SCs and PV+ INs may occupy discrete zones that correspond to reported grid cell clusters (*Gu et al., 2018*; *Heys et al., 2014*), while indirect interactions may operate at the scale of grid cell modules.

### Assessment of common input to postsynaptic neurons

Functional investigation of synaptic connectivity, including within MEC (*Couey et al., 2013*; *Dhillon and Jones, 2000*; *Fuchs et al., 2016*; *Pastoll et al., 2013*; *Winterer et al., 2017*), typically relies on simultaneous recordings in which pre- and postsynaptic neurons are identified but are usually within 150 μm of one another. Local optical activation of unidentified neurons has also been used to infer connectivity principles but with a focus on responses of single postsynaptic neurons (*Beed et al., 2013*; *Beed et al., 2010*). In contrast, our primary goal was to address the coordination of downstream neurons by genetically defined upstream neurons. To do so, we introduced new strategies that are potentially applicable to many brain areas (*Figures 2 and 5*). By recording from pairs of PV+ INs (or SCs), and optogenetically activating SCs (or PV+ INs) we were able to assess the extent and organization of projections from one population to the other. We find that with either wide-field

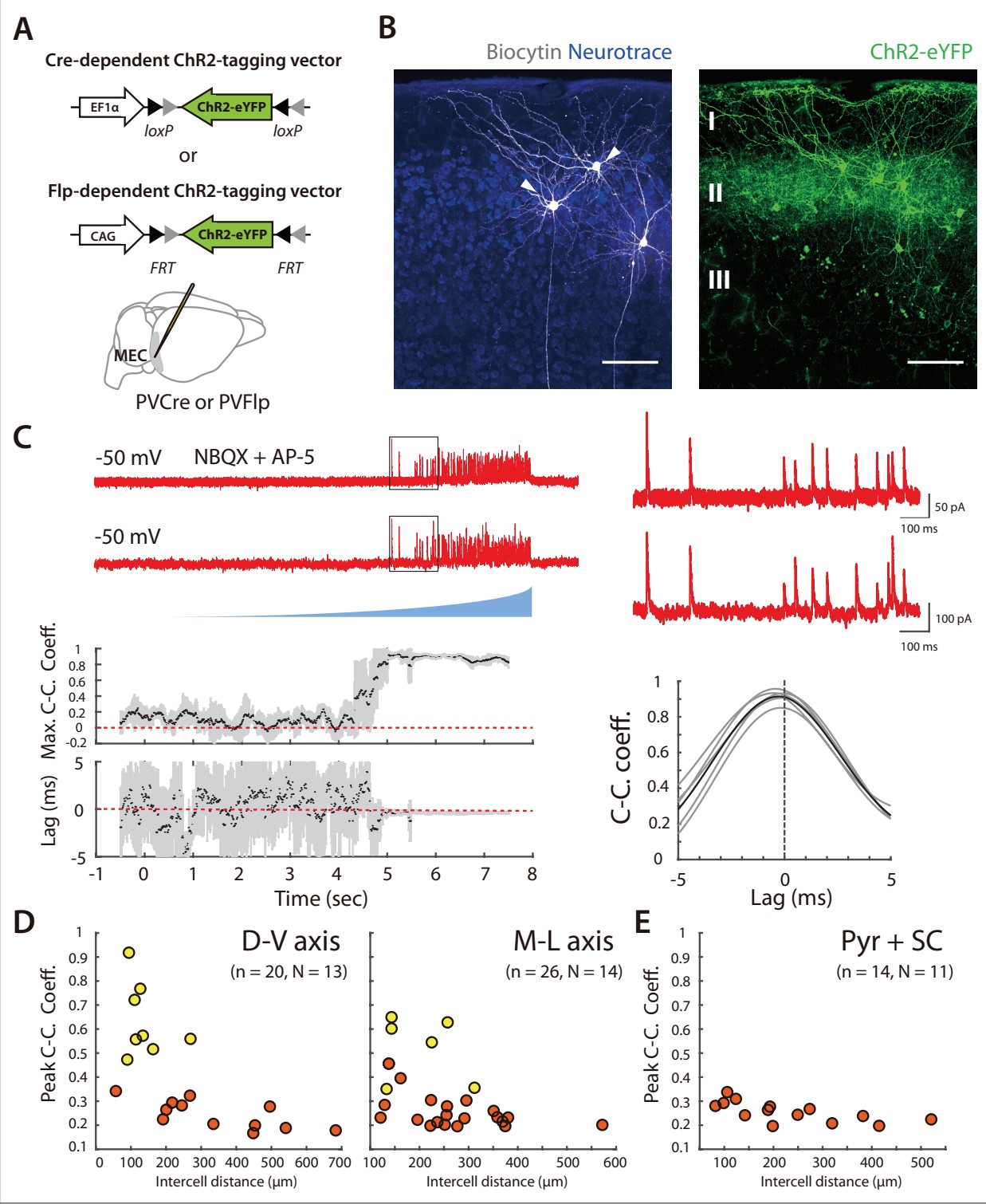

**Figure 7.** Organization of correlated inputs from PV+ interneurons to SCs. (**A**) Targeting of ChR2 to PV+ INs used Cre- or FLP-dependent AAVs injected into the MEC of PVCre or PVFlp mice. (**B**) Whole-cell recordings were targeted to pairs of principal cells in layer 2 of the MEC (left, arrows) within the region of viral infection (right). Scale bar = 100 μm. (**C**) Example traces of simultaneously recorded IPSCs from the pair of SCs shown in (**B**) in response to ramp-modulated light stimulation. (**D**) Peak cross-correlation coefficient as a function of intercell distance along the dorsal-ventral axis (left panel; adjusted $R^2$=0.41, p=0.0014, $F_{1,18}$ = 14.3) and medial-lateral axis (right panel; adjusted $R^2$=0.17, = 0.02, $F_{1,24}$ = 6.0) for simultaneously recorded SC pairs. Pairs of cells meeting the criteria (peak cross-correlation coefficients >0.35 and trough values of variance of lags <0.2) are indicated in yellow, else in

*Figure 7 continued on next page*

*Figure 7 continued*

orange. (**E**) Peak cross-correlation coefficient as a function of intercell distance for all the simultaneously recorded SC and pyramidal cell pairs (D-V and M-L axes).

The online version of this article includes the following figure supplement(s) for figure 7:

**Figure supplement 1.** Distribution of SCs receiving correlated input from PV+ INs.

or focussed optical activation of upstream neurons defined by genetically targeted expression of ChR2, it is possible to identify pairs of downstream neurons that receive common input. While with either strategy it is possible that weak connections are not detected, results from the two approaches corroborate one another showing similar location-dependence of functional synaptic coordination. Our focal optogenetic stimulation approach also offers insight into the spatial organization of presynaptic neuronal populations, with the advantage, compared to focal glutamate uncaging previously used to investigate connectivity in the MEC (*Beed et al., 2013*; *Beed et al., 2010*), that the identity of the presynaptic cell population is genetically defined.

Our approach to identifying common inputs to pairs of simultaneously recorded neurons is conceptually similar to classic investigations of circuits in visual cortex, which used focal glutamate uncaging to activate upstream neurons while recording from pairs of postsynaptic neurons (*Yoshimura et al., 2005*; *Yoshimura and Callaway, 2005*). An advantage of optogenetic activation is that we were able to address connectivity between genetically defined populations of pre- and postsynaptic neurons. A possible concern with focal single photon optogenetic activation is that activation has limited spatial specificity (*Anastasiades et al., 2018*). This means we are not able to identify the individual activated presynaptic neurons, but strict somatic localisation is not necessary for our conclusions based on asynchronous activation of presynaptic neurons. Nevertheless, adoption of soma-targeted opsins would be a useful extension to our approach by facilitating mapping of the activated presynaptic neurons.

Our widefield (*Figures 2–4 and 7*) and focal activation approaches (*Figure 6*) both rely on the fact that neurons expressing ChR2 have different thresholds for initiation of spikes, and that their spike latency is variable from trial to trial. Thus, responses in downstream cells that are highly correlated may either originate from the same presynaptic neuron, or from presynaptic neurons with activity coordinated through multisynaptic pathways. It is important to note here that the absence of correlated inputs in one slice plane does not rule out the possibility that the same cell pair receives common inputs in a different plane, as these inputs would most likely not be activated if the cell bodies of the presynaptic neuron were removed by slicing. Multisynaptic coordination would require ChR2 expressing SCs to activate other neuronal populations. Such intermediary populations are unlikely to include other SCs or pyramidal cells as inputs to both populations from SCs are weak or absent (*Couey et al., 2013*; *Dhillon and Jones, 2000*; *Fuchs et al., 2016*; *Pastoll et al., 2013*; *Winterer et al., 2017*). Consistent with this, we did not observe synchronous activation of optogenetically activated SCs (*Figure 2—figure supplement 1*). Instead, the sensitivity of correlated inputs to distant PV+ INs to block of fast GABAergic transmission is consistent with multisynaptic coordination involving interneuron populations. By contrast, correlated responses of nearby PV+ INs appear unlikely to involve multisynaptic coordination as they were maintained during block of fast GABAergic transmission and were observed following focal stimulation, which because it is sparse is unlikely to recruit multisynaptic pathways. Responses of SCs to ramping activation are also unlikely to involve multisynaptic coordination as the activated PV+ INs would inhibit rather than drive activity of their downstream targets.

## Spatial extent and functional organization of L2 circuits

Our results demonstrate that activation of SCs leads to common excitatory synaptic input to pairs of PV+ INs, and activation of PV+ INs leads to common synaptic input to pairs of SCs. Several features of this shared input may be relevant to network computations in L2 of the MEC. (1) Correlation coefficients indicative of common input in each direction were always less than 1 and usually in the region of 0.4–0.6 (*Figures 3, 5 and 7*). Thus connectivity in each direction may be incomplete rather than all to all, and the input to any pair of neurons is likely only to be partially overlapping (*Figure 2F*). The incompleteness may in part result from loss of some inputs by tissue slicing. However, the fact that axons were well preserved and typically extended beyond the range of functional correlations, while many cell pairs that did not receive correlated input were relatively close to one another and

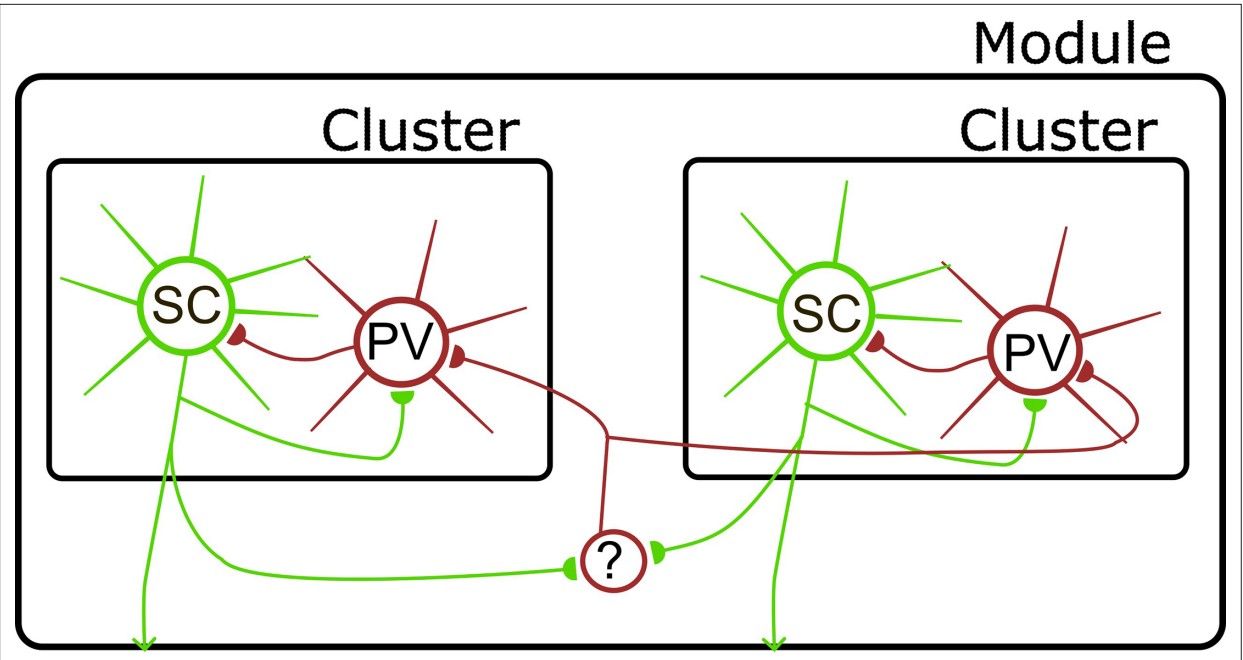

**Figure 8.** Model for organization of circuits in L2 of the MEC. Direct interactions between SCs and PV+ INs define local clusters (see also *Couey et al., 2013*; *Dhillon and Jones, 2000*; *Fuchs et al., 2016*; *Pastoll et al., 2012a*; *Winterer et al., 2017*). Interactions between clusters are mediated via inhibitory interneurons (?), which could include somatostatin positive interneuron in layer 2 (*Fernandez et al., 2022*) or interneurons in layer 1 (*Shi et al., 2023*; *Vandrey et al., 2022*). Circles here represent populations of neurons, with connections between each population being incomplete rather than all to all.

had overlapping dendritic fields, argues against tissue slicing being a major contributor to incompleteness. (2) Local interactions through direct electrical synapses appear only able to coordinate PV+ INs with short separations (<100 μm) relative to the extent of functional coordination, while synaptic interactions between PV+ INs were not detected (*Figure 3—figure supplement 2*). (3) Many nearby cells don't appear to share common input (*Figures 3, 5 and 7*), suggesting they are part of distinct networks, or are separated within the connectivity space of the same network. (4) Indirect inhibitory interactions can boost coordination for inputs in the dorsoventral plane, increasing the spatial extent of relatively strong correlations (*Figure 4*). (5) The organization of slice locations that activate inputs from SCs to PV+ INs is relatively similar for neurons that receive common input, can be anticorrelated for neurons that do not, and can often show discrete borders (*Figure 5*). This is consistent with a modular organization within layer 2. (6) Pairs of SCs and pyramidal cells in L2 do not generate correlated responses when PV+ INs are activated. This is consistent with SCs and pyramidal cells receiving inhibitory input from distinct interneuron populations (*Fuchs et al., 2016*; *Varga et al., 2010*) (but see *Armstrong et al., 2016*).

### Ideas and speculation

Based on our results here, together with previous investigations of MEC connectivity using dual patch-clamp and optogenetic approaches, we suggest a tentative circuit model for L2 of the MEC in which interactions between SCs and PV+ INs arise at several spatial scales (*Figure 8*). First, PV+ INs separated by <100 μm interact through gap junction connections (see also *Fernandez et al., 2022*). Second, populations of interacting PV+ INs and SCs separated by up to ~250 μm form local modules defined by the common inputs between each cell population (see also *Couey et al., 2013*; *Dhillon and Jones, 2000*; *Fuchs et al., 2016*; *Pastoll et al., 2013*; *Winterer et al., 2017*). Third, longer range output from SCs to non-PV INs may support coordination between modules. Candidates for this coordination could include interneurons in L1 of the MEC (*Vandrey et al., 2022*), which can have extensive intra-laminar axons (*Shi et al., 2023*), or somatostatin-positive interneurons in layer 2 (*Fernandez et al., 2022*). We suggest this model as a starting point for further investigation of MEC connectivity.

What functional predictions follow from this model? It is of interest to compare the spatial scales for functional connectivity identified here with local correlations in grid cell activity (*Heys et al., 2014*) and with the organization of grid cell networks into modules composed of local clusters (*Gu et al., 2018*). In principle, local shared inputs from SCs to PV+ INs, and vice versa, could form a substrate for computation within grid cell clusters. Given the continuous attractor dynamics of grid cell populations observed experimentally (*Barry et al., 2007*; *Gardner et al., 2021*; *Yoon et al., 2013*), such connectivity could support emergence of activity bumps through excitatory-inhibitory interactions (*Guanella et al., 2007*; *Pastoll et al., 2013*; *Shipston-Sharman et al., 2016*; *Widloski et al., 2018*; *Widloski and Fiete, 2014*). The incomplete connectivity between local SCs and PV+ INs is consistent with structured connections expected within networks with continuous attractor dynamics (*Pastoll et al., 2013*; *Shipston-Sharman et al., 2016*). In this framework, intra-modular coordination could result from interactions between clusters mediated by non-PV interneurons. Discrete and incomplete local connectivity could also be consistent with models for grid firing that do not require synaptic interactions between SCs or that focus on pyramidal neurons (e.g. *Burgess et al., 2007*; *Grossberg and Pilly, 2012*; *Kropff and Treves, 2008*). However, these models do not offer any explanation for the structured local circuitry and also do not account for the experimental evidence for continuous attractor dynamics of grid cell populations. Addressing these speculations will require strategies for improving throughput for circuit interrogation, for example through application of all optical approaches (*Hochbaum et al., 2014*), and for mapping of connectivity onto the in vivo firing properties of identified neurons (*Ko et al., 2011*).

## Methods

### Animals

Experiments were carried out under a project license granted by the UK Home Office and according to the guidelines laid down by the University of Edinburgh's Animal Welfare Committee, the policies of the UK Animals (Scientific Procedures) Act 1986 and European Directive 2010/62/EU on the protection of animals used for experimental purposes. Male (43, 50%) and female (43, 50%) mice were both used. Mice were randomly assigned to experimental groups, and up to six mice were kept in a single cage. The following transgenic mouse lines were obtained from Jackson labs: PVFlp mice (B6.Cg-Pvalb-tm4.1(FlPo)Hze/J, Jackson Lab stock number 022730), which express Flp-recombinase downstream from the PV promoter; PVCre mice (B6;129P2-Pvalbtm1(cre)Arbr/J, Jackson Lab stock number 008069), which express Cre-recombinase downstream from the PV (Pvalb) promoter; Sim1Cre mice (strain name: Tg(Sim1cre)KJ21Gsat/ Mmucd, Mutant Mouse Resource and Research Centres stock number 034614-UCD), which express Cre-recombinase downstream from the Sim1 promoter.

### Injection of viruses

The following AAV vectors were used for channelrhodopsin-2 expression: pAAV-EF1a-double floxed-hChR2 (H134R)-EYFP-WPRE-HGHpA (Addgene plasmid #20298), pAAV-EF1a-double floxed-hChR2 (H134R)-mCherry-WPRE-HGHpA (Addgene plasmid #20297). Flp recombinase dependent AAV vectors were previously constructed in house (see *Tsoi et al., 2022*). All AAV preps were generated in house as previously described (*Murray et al., 2011*; *Sürmeli et al., 2015*) and were titered by qPCR following expression in HEK cells (>$10^{12}$ GC/ml).

For experiments using multiple viruses they were mixed in equal volumes and injected simultaneously. Viruses were injected into either PvFlp x Sim1Cre mice, PVFlp or PVCre mice. Mice were 5–8 weeks old at the time of injection. Injections targeted the medial entorhinal cortex following procedures described previously (*Sürmeli et al., 2015*). Mice were anesthetized with isoflurane throughout the surgical procedures. After mounting to a stereotaxic frame, a craniotomy was made at 3.4 mm lateral and 0.6 mm posterior to bregma in the right hemisphere. Viruses were injected using a calibrated micropipette (Drummond Scientific) pulled to a tip diameter of approximately 7–10 μm. For experiments using sagittal brain slices, three injections were made from the same surface coordinates with angles relative to the vertical axis of 8°, 10°, and 12°. For experiments using horizontal brain slices additional injections were carried out at 3.3 mm and 3.5 mm lateral and 0.6 mm posterior to bregma, with an angle of 10° relative to vertical. The depth for each injection was determined by advancing

the pipette until a slight bending was observed and then withdrawing the pipette 0.2 mm before injecting. 400 nl of viral mixture was injected at each position at a rate of approximately 20 nl / s.

## Electrophysiological recordings

Sagittal or horizontal brain slices of thickness 300 or 400 μm were prepared from the right hemisphere 5–7 weeks after viral injections as previously described (*Garden et al., 2008*; *Pastoll et al., 2012b*). For each question we aimed to obtain data from a minimum of eight mice, with final sample sizes determined by the maximum number of recordings that could be made given the available numbers of mice. Mice were sacrificed by cervical dislocation and their brains carefully removed. Slices were prepared in cutting solution at 4 °C with composition (mM): NaCl 86, NaH$_2$PO4 1.2, KCl 2.5, NaHCO$_3$ 25, Glucose 20, Sucrose 75, CaCl$_2$ 0.5, MgCl$_2$ 7. For recordings slices were placed in a chamber constantly perfused with oxygenated (95% O$_2$/5% CO$_2$) artificial cerebrospinal fluid maintained at 33–34°C containing composition (mM): NaCl 124, NaH$_2$PO$_4$ 1.2, KCl 2.5, NaHCO$_3$ 25, Glucose 20, CaCl$_2$ 2, MgCl$_2$ 1. Slices were visually inspected with epifluorescence to confirm specific expression of ChR2 in superficial layers of the MEC. Animals in which ChR2-expressing cell bodies were not restricted to layer 2 of the MEC were discarded from analysis.

Whole-cell recordings were performed using glass micropipettes pulled on a Sutter P-97 Micropipette Puller. For standard whole-cell recordings the intracellular solution had composition (mM): K-Gluconate 130; KCl 10, HEPES 10, MgCl$_2$ 2, EGTA 0.1, Na$_2$ATP 2, Na$_2$GTP 0.3, Na-Phosphocreatine 10 and biocytin 0.5%. The pipette resistance measured in the recording bath was 4–6 MΩ.

Series resistance was measured either by balancing the bridge in the current-clamp recording configuration or through compensation of capacitance transients in the voltage-clamp configuration, and was ≤25 MΩ (range 6.9–25 MΩ) for voltage-clamp experiments and <30 MΩ (range 13.3–27.7 MΩ) for current-clamp experiments. Series resistance in voltage-clamp recordings was compensated up to 70%. Voltage-clamp recordings were performed with a holding potential of –70 mV unless otherwise indicated. Data were acquired and filtered at 6 kHz using a Multiclamp 700B Amplifier (Axon Instruments) and digitized at 20 kHz. All data were acquired using Axograph software.

To block GABA$_A$ receptor-mediated inhibitory synaptic transmission 20 μM GABAzine (Hello Bio) was added to the standard ACSF perfusing the slice. Experiments recording isolated inhibitory postsynaptic currents used ACSF including 5 μM NBQX (Hello Bio) and 50 μM D-AP5 (Hello Bio) and a modified intracellular solution with composition (mM): KCl 140, HEPES 10, MgCl$_2$ 2, EGTA 0.1, Na$_2$ATP 2, Na$_2$GTP 0.3, Na-Phosphocreatine 10 and biocytin 0.5%. Stellate cells were identified by their characteristic sag potentials in response to negative and positive current steps and by their prominent afterhyperpolarization (*Pastoll et al., 2012a*). To facilitate recovery of the axonal morphology recordings were maintained for at least 1 hr (see also *Jiang et al., 2015*).

## Neuronal reconstruction

To reveal biocytin labeling of the axonal and dendritic morphology of the recorded neurons we adopted previously described procedures (*Jiang et al., 2015*). Briefly, following recording slices were fixed overnight at 4 °C in 4% paraformaldehyde. They were then washed in phosphate buffered saline (PBS), incubated overnight with Streptavidin-Alexa 647 (1:500 in PBS), washed in PBS containing 0.3% triton, dried and mounted using Mowiol 4–88 (Aldrich). Identification of stellate cells and pyramidal cells was further confirmed post-hoc by the multiple dendritic processes in the former and the prominent apical dendrites in the latter. Cell data with incomplete biocytin-filled morphology were discarded from analysis. Stellate cells were excluded if the axon was cut prior to leaving the entorhinal cortex. Sample sizes were determined by the number of successful reconstructions that could be obtained from biocytin filled neurons.

Biocytin-filled cells were manually reconstructed using Neurolucida 360 software (MBF Bioscience) and analyzed using Neurolucida Explorer (MBF Bioscience). To correct X and Y shrinkage, a ×4 magnification image from the electrophysiology rig was compared to a 4 x confocal image using a 250 μm grid scaled for the electrophysiology rig. Distances of interest (e.g. distance from the cell to the dorsal border of the MEC) were scaled from the 4 x image to the 4 x confocal image (*Pastoll et al., 2012a*). To correct Z shrinkage the depth of the imaged fixed slice was rescaled to the depth of the slice during recording.

## Photo-stimulation

For widefield activation of ChR2 we used a 470 nm LED (Thorlabs) attached to the epifluorescence port of the experimental microscope (Olympus BX-51), and focused onto the slice chamber using a 40 x objective (Olympus). The irradiance of the LED was controlled by custom analogue voltage commands. The maximal intensity used in this study was ~22 mW/mm². The region of neurons activated by the LED had a radius of approximately 100 µm and depth greater than 100 µm. Before evaluating responses to asynchronous activation of presynaptic neurons, we tested each cell pair for functional ChR2 driven postsynaptic responses using a 3ms duration square pulse of intensity ~22 mW/mm² delivered 3 times. We included cell pairs for analysis when both cells demonstrated suprathreshold membrane potential responses in current-clamp mode or amplitude of EPSCs/IPSCs >100 pA in voltage-clamp mode. To asynchronously activate ChR2 expressing neurons, we used an 8 s duration ramp-modulated light stimulus (~10 mW/mm² final intensity) repeated 5 times and with the field of view centered approximately at the equal point between two simultaneously recorded cells.

For focal activation of ChR2, laser light was generated using a 100 mW 473 nm diode laser (RappOpto; Hamburg, Germany), integrated into the light path of the microscope and focused onto the slice using a 40 x objective. A neutral density filter with 1% transmission rate was used to attenuate the laser beam. In the focal plane of the slice the diameter of the stimulation site was ~1 µm. Stimulation sites were organized in a 7x10 grid with adjacent sites spaced 20 µm apart along each row and column, giving an overall scanning area of 120 µm x 180 µm. The longer scanning axis was parallel to the dorsal-ventral axis in sagittal slices and the medial-lateral axis in horizontal slices. The center of the laser scanning area was focused on the region where ChR2-expressing cell bodies were most abundant. In cases where the viral expression between the pair of recorded cells was homogenous, the objective was focused approximately at the equal point between the two recorded cells. In the laser scanning protocol, a 1 ms light pulse was applied to one location at a time. Five locations were stimulated per second in a serial, pseudorandom order, with 5 s breaks after every 15 s of laser scanning. The laser scanning was repeated 10 times for each laser power. Data were sampled at a range of laser powers, from the intensity that gives minimum light-evoked responses to the minimum laser intensity that renders light-evoked responses from all the 70 scanning sites. The step size of linear increment of laser power was 10 or 100 µW, according to the range of light responsiveness of recorded cells. The trials in which the baseline was fluctuating were excluded from analysis.

## Data analysis, simulations, and statistics

All data were analyzed using built in and custom written routines in IGOR pro (Wavemetrics), Axograph, MATLAB (Statistics and Machine Learning Toolbox; Mathworks) and R (R Core Team).

### Simulation of convergent synaptic input

The goal of our simulations was to assess if cross-correlations between somatic membrane currents recorded from pairs of neurons could be used to establish whether they receive shared input from the same pre-synaptic neuron. While this should be obvious if neurons exclusively receive shared input, we wanted to establish whether shared input is detectable when each neuron also receives independent inputs of similar frequency and amplitude to the shared input. Because we considered experimental conditions in which neurons are voltage-clamped, and therefore capacitance currents are zero and voltage-gated currents are stationary, we simulate only the synaptic currents resulting from activation of populations of presynaptic SCs (*Figure 2*). To generate times for activation of the presynaptic inputs, we first recorded responses of ChR2 expressing SCs to ramping light stimuli and extracted the spike times. We then convolved the spike times with simulated postsynaptic EPSC waveforms, approximating those recorded from PV+ INs, described using the following functions:

$$g = g_{max} * \left(t - t_{spike}\right)/tau * e^{\left[-\left(t - t_{spike} - tau\right)/tau\right]}$$

$$i = g * \left(v - e\right)$$

where $g$ is the conductance, $g_{max}$ is the maximum conductance (set to 1 µS), $tau$ is the time to peak of the conductance change (1ms), $i$ is the synaptic current, $v$ is the resting membrane potential (–70 mV) and $e$ is the equilibrium potential for the transmitter-gated event (0 mV). This resulted in simulated

postsynaptic current responses to each presynaptic spike train. To capture trial-to-trial variability in timing we computed current responses for multiple ramping responses of each SC.

Because experiments were conducted in the voltage-clamp configuration, we simulate responses to activation of multiple presynaptic SCs by summing the individual current traces generated from spike trains of different experimentally recorded SCs. In each simulation, two postsynaptic neurons, equivalent to the experimentally voltage-clamped PV+ INs, received input from 8 SCs, while the number of shared presynaptic inputs varied systematically between simulations.

## Analyses of electrophysiological data

To evaluate potential shared input to pairs of neurons on the basis of responses to ramp-like optical activation of presynaptic neurons, the recorded postsynaptic membrane currents were first bandpass filtered between 10 and 120 Hz. The cross-correlation coefficients between filtered membrane currents from each pair of simultaneously recorded neurons were then calculated using a 250 ms time window that was shifted at 10ms intervals across the duration of the response to the ramp stimulus. The maximum cross-correlation coefficient was calculated by averaging and smoothing cross-correlation coefficients over five trials, and extracting the maximal value during photo-stimulation. The minimum variance of time lags was calculated by averaging and smoothing the standard deviation of time lags over 5 trials, extracting and squaring the minimal value during the time series of photo-stimulation.

To evaluate shared input on the basis of experiments with focal optical activation of presynaptic neurons, EPSCs were first detected from the recorded postsynaptic currents using the template matching function implemented in Axograph. We defined a template EPSC from experimental data, and then set the algorithm to detect the shape of EPSC events that matches the template including the events that overlap partially in time. The signal-to-noise ratio for detecting events was set to 2.5. Correlation probabilities were calculated using previously described procedures (*Yoshimura et al., 2005*). Cross-correlograms were calculated for each pair of stimulation locations using 4 ms histogram bins, with the central bin spanning 0±2ms. The cross-correlograms for each scanning location were summed up. The same data and approach, but with stimulus labels shifted, was used to generate shifted cross-correlograms. The value in the central bin of the shifted cross-correlogram was subtracted from that of the unshifted cross-correlogram. The result was then normalized by dividing by the total number of evoked EPSCs, to give the correlation probability.

## Anatomical reconstructions

The densities of reconstructed axons and dendrites were calculated using the TREES toolbox (*Cuntz et al., 2011*). The size of voxels was 5 μm in the calculation of densities of cellular processes. We varied the distance between two postsynaptic cells and their locations relative to the presynaptic cell along either dorsoventral or mediolateral axes in each simulation. An estimate of axodendritic overlapping was measured by multiplying voxel by voxel densities of presynaptic and postsynaptic neuronal processes and summing across all voxels. We normalized this value with the value obtained by setting the soma of three neurons to the same coordinates, which gives a ratio of overlapping. To specifically evaluate the axodendritic overlapping from one presynaptic neurons onto two postsynaptic neurons, we only consider the cases where the axodendritic densities of both pre- and postsynaptic pairs are positive values, otherwise we set the ratio of overlapping to be zero.

## Statistics

For group comparisons where the distribution of data points was approximately normal we used standard parametric statistics. Differences in axon and dendritic length between layers and slice plane were evaluated with two-way repeated measures ANOVA, with slice plane and layer as the group variables and repeated measures used because the same neuron contributes measures to each layer (*Figures 1E and 6E*). The relative extent of axons and dendrites were compared using paired t-tests (*Figures 1G and 6G*). Comparisons of correlograms used ANOVA with time bin and whether the correlogram was directly obtained from the data or from shuffled data as group variables (*Figure 2—figure supplement 1*). Relationships between variables were evaluated by fitting linear models (*Figures 1F, 3E, F, 5F, H, 6F, 7D*). While linear models will not detect complex relationships in the data we chose them here to avoid the possibility of overfitting that may arise from use of more complex models.

Analysis code and datasets are available from https://datashare.ed.ac.uk/handle/10283/8873.

## Additional information

### Funding

| Funder | Grant reference number | Author |
|---|---|---|
| Biotechnology and Biological Sciences Research Council | BB/M025454/1 | Matthew F Nolan |
| Wellcome Trust | 10.35802/200855 | Matthew F Nolan |
| Simons Initiative for the Developing Brain | | Derek LF Garden Matthew F Nolan |

The funders had no role in study design, data collection and interpretation, or the decision to submit the work for publication. For the purpose of Open Access, the authors have applied a CC BY public copyright license to any Author Accepted Manuscript version arising from this submission.

### Author contributions

Li-Wen Huang, Conceptualization, Formal analysis, Investigation, Visualization, Methodology, Writing – review and editing; Derek LF Garden, Data curation, Formal analysis, Investigation, Visualization, Methodology, Writing – review and editing; Christina McClure, Methodology, Writing – review and editing; Matthew F Nolan, Conceptualization, Formal analysis, Supervision, Funding acquisition, Visualization, Writing – original draft, Project administration, Writing – review and editing

### Author ORCIDs

Derek LF Garden ⬤ https://orcid.org/0000-0003-3336-3791
Matthew F Nolan ⬤ https://orcid.org/0000-0003-1062-6501

### Ethics

Experiments were carried out under a project license granted by the UK Home Office (PP5105514) and according to the guidelines laid down by the University of Edinburgh's Animal Welfare Committee, the policies of the UK Animals (Scientific Procedures) Act 1986 and European Directive 2010/62/EU on the protection of animals used for experimental purposes.

Reviewer #1 (Public review): https://doi.org/10.7554/eLife.92854.4.sa1
Reviewer #3 (Public review): https://doi.org/10.7554/eLife.92854.4.sa2
Author response https://doi.org/10.7554/eLife.92854.4.sa3

## Additional files

### Supplementary files
• MDAR checklist

### Data availability

Data are deposited in the University of Edinburgh DataShare repository (https://doi.org/10.7488/ds/7813). Neuronal reconstructions are also available from the Neuromorpho repository (https://neuromorpho.org). To download neuronal reconstructions search for 'Nolan' lab under 'Browse by archive' (https://neuromorpho.org/bylab.jsp). Neurons described in this study have identifiers beginning with 'Huang'.

The following dataset was generated:

| Author(s) | Year | Dataset title | Dataset URL | Database and Identifier |
|---|---|---|---|---|
| Huang L-W, Garden D, Nolan M | 2024 | Synaptic interactions between stellate cells and parvalbumin interneurons in layer 2 of the medial entorhinal cortex are organized at the scale of grid cell clusters | https://doi.org/10.7488/ds/7813 | Edinburgh DataShare, 10.7488/ds/7813 |

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
