## [Editor Report · eLife Assessment]

In this **compelling** study, the authors examine the interactions between stellate cells and PV+ interneurons in the medial entorhinal cortex. Huang et al. focus on the spatial distribution of synaptic inputs and demonstrate that closely located neuron pairs receive common inputs, suggesting a structured functional organization in the entorhinal cortex. Advanced dual whole-cell patch recordings further reveal patterns of postsynaptic activation, indicating intensive interactions within clusters of these neurons, with weaker interactions between clusters. These findings offer significant insights into the functional dynamics of the entorhinal cortex and the circuit mechanisms that shape grid cell activity. This study is **important** not only for the field of MEC and grid cells, but also for broader fields of continuous attractor networks and neural circuits.

---

## [Referee Report · Reviewer #1 (Public review)]

Summary:

The circuit mechanism underlying the formation of grid cell activity and the organization of grid cells in the medial entorhinal cortex (MEC) is still unclear. To understand the mechanism, the current study investigated synaptic interactions between stellate cells (SC) and PV+ interneurons (IN) in layer 2 of the MEC by combing optogenetic activations and paired patch-clamp recordings. The results convincingly demonstrated highly structured interactions between these neurons: specific and direct excitatory-inhibitory interactions existed at the scale of grid cell phase clusters, and indirect interactions occurred at the scale of grid modules.

Strengths:

Overall, the manuscript is very well written, the approaches used are clever, and the data were thoroughly analyzed. The study conveyed important information towards understanding the circuit mechanism that shapes grid cell activity. It is important not only for the field of MEC and grid cells, but also for broader fields of continuous attractor network and neural circuit.

Weaknesses:

The study largely relies on the fact that ramp-like wide field optogenetic stimulation and focal optogenetic activation both drove asynchronous action potentials in SCs, and therefore, if a pair of PV+ INs exhibited correlated activity, they should receive common inputs. While the asynchronization of action potentials during ramp-like wide field optogenetics was shown in Figure 2 Figure Supplement 1, the asynchronization during focal optogenetic activation was not confirmed in the current experimental setting. More data and statistical analysis in this aspect would strengthen the foundation of this study.

---

## [Referee Report · Reviewer #3 (Public review)]

Summary:

This paper presents convincing data from technically demanding dual whole cell patch recordings of stellate cells in medial entorhinal cortex slice preparations during optogenetic stimulation of PV+ interneurons. The authors show that the patterns of postsynaptic activation are consistent with dual recorded cell close to each other receiving shared inhibitory input and sending excitatory connections back to the same PV neurons, supporting a circuitry in which clusters of stellate cells and PV+ IN interact with each other with much weaker interactions between clusters. These data are important to our understanding of the dynamics of functional cell responses in the entorhinal cortex. The experiments and analysis are quite complex and would benefit from some revisions to enhance clarity.

Strengths:

These are technically demanding experiments, but the authors show quite convincing differences in the correlated response of cell pairs that are close to each other in contrast to an absence of correlation in other cell pairs at a range of relative distances. This supports their main point of demonstrating anatomical clusters of cells receiving shared inhibitory input.

Weaknesses:

The overall technique is complex, but the authors have made every effort to present this in a clear manner. In addition, due to this being a slice preparation they cannot directly relate the inhibitory interactions to the functional properties of grid cells which was possible in a complementary approach using 2-photon in vivo imaging by Heys, Rangarajan and Dombeck, 2014.

---

## [Author Response]

The following is the authors’ response to the previous reviews.

**Reviewer #1 (Public Review):**
Overall, the manuscript is very well written, the approaches used are clever, and the data were thoroughly analyzed. The study conveyed important information for understanding the circuit mechanism that shapes grid cell activity. It is important not only for the field of MEC and grid cells, but also for broader fields of continuous attractor networks and neural circuits.

We appreciate the positive comments.

(1) The study largely relies on the fact that ramp-like wide-field optogenetic stimulation and focal optogenetic activation both drove asynchronous action potentials in SCs, and therefore, if a pair of PV+ INs exhibited correlated activity, they should receive common inputs. However, it is unclear what criteria/thresholds were used to determine the level of activity asynchronization, and under these criteria, what percentage of cells actually showed synchronized or less asynchronized activity. A notable percentage of synchronized or less asynchronized SCs could complicate the results, i.e., PV+ INs with correlated activity could receive inputs from different SCs (different inputs), which had synchronized activity. More detailed information/statistics about the asynchronization of SC activity is necessary for interpreting the results.

The percentage of SCs that show synchronised activity during ramping optogenetic activation is zero. To make this clear we've added new quantification to the analyses of simultaneously activated SCs in Figure 2, Figure Supplement 1. This includes confidence intervals for the correlograms and statistical comparisons of the correlograms to shuffled data from each pair of neurons. We also validate our statistical analysis strategy by showing that it successfully identifies autocorrelation peaks for the same cells.

Synchronisation during focal optogenetic activation is also expected to be zero. We did not commit resources to experiments to directly test this for focal stimulation because we had already tested the possibility with ramping stimuli discussed above, and because the established biophysics of local SC circuits is such that synchronised activity during selective activation of SCs is unlikely. In particular, because direct excitatory connections between SCs are either rare or absent (Fuchs et al. 2016; Couey et al. 2013; Pastoll et al. 2013; Winterer et al. 2017), and when detected have small amplitude (Winterer et al. 2017), no mechanism exists that could drive synchronisation. The absence of coordination in responses to ramping stimuli quantified above is consistent with this conclusion.

(2) The hypothesis about the "direct excitatory-inhibitory" synaptic interactions is made based on the GABAzine experiments in Figure 4. In the Figure 8 diagram, the direct interaction is illustrated between PV+ INs and SCs. However, the evidence supporting this "direct interaction" between these two cell types is missing. Is it possible that pyramidal cells are also involved in this interaction? Some pieces of evidence or discussions are necessary to further support the "direction interaction".

We were insufficiently clear in our previous attempts to ground these interpretations in the context of previous work. The hypothesis about "direct excitatory-inhibitory" interactions wasn't made solely on the basis of Figure 4, but from multiple previous studies that directly demonstrate these interactions (e.g. Fuchs et al. 2016; Couey et al. 2013; Pastoll et al. 2013). Similarly, the diagram in Figure 8 doesn't only reflect the conclusions of the present study but integrates work from these and other previous studies.

A possible role for pyramidal cells in coordination would require that they can be driven to fire action potentials by input from SCs. However, SCs appear not to connect to pyramidal cells (0/126 tested connections in Winterer et al. 2017). Thus, this possibility is inconsistent with the previously published data.

To make these points clearer we have added additional discussion and citations to the results (p 5), discussion (p 11) and legend to Figure 8.

**Reviewer #2 (Public Review):**
In this study, Huang et al. employed optogenetic stimulation alongside paired whole-cell recordings in genetically defined neuron populations of the medial entorhinal cortex to examine the spatial distribution of synaptic inputs and the functional-anatomical structure of the MEC. They specifically studied the spatial distribution of synaptic inputs from parvalbumin-expressing interneurons to pairs of excitatory stellate cells. Additionally, they explored the spatial distribution of synaptic inputs to pairs of PV INs. Their results indicate that both pairs of SCs and PV INs generally receive common input when their relative somata are within 200-300 ums of each other. The research is intriguing, with controlled and systematic methodologies. There are interesting takeaways based on the implications of this work to grid cell network organization in MEC.

We appreciate the positive comments.

(1) Results indicate that in brain slices, nearby cells typically share a higher degree of common input. However, some proximate cells lack this shared input. The authors interpret these findings as: "Many cells in close proximity don't seem to share common input, as illustrated in Figures 3, 5, and 7. This implies that these cells might belong to separate networks or exist in distinct regions of the connectivity space within the same network.".Every slice orientation could have potentially shared inputs from an orthogonal direction that are unavoidably eliminated. For instance, in a horizontal section, shared inputs to two SCs might be situated either dorsally or ventrally from the horizontal cut, and thus removed during slicing. Given the synaptic connection distributions observed within each intact orientation, and considering these distributions appear symmetrically in both horizontal and sagittal sections, the authors should be equipped to estimate the potential number of inputs absent due to sectioning in the orthogonal direction. How might this estimate influence the findings, especially those indicating that many close neurons don't have shared inputs?

We appreciate the suggestion, however systematically generating estimates that account in full for the relative position of the postsynaptic neurons, for variation in the organisation of their dendritic fields and for unknowns such as the location and number of synaptic contacts made, quickly leads to a large potential parameter space, while not advancing our understanding beyond qualitative assessment of the raw data.

Given this, we make the following comments:

'We note that the absence of correlated inputs in one slice plane does not rule out the possibility that the same cell pair receives common inputs in a different plane, as these inputs would most likely not be activated if the cell bodies of the presynaptic neuron were removed by slicing.' (p10) and:

'The incompleteness may in part result from loss of some inputs by tissue slicing. However, the fact that axons were well preserved and typically extended beyond the range of functional correlations, while many cell pairs that did not receive correlated input were relatively close to one another and had overlapping dendritic fields, argues against tissue slicing being a major contributor to incompleteness.' (p10).

(2) The study examines correlations during various light-intensity phases of the ramp stimuli. One wonders if the spatial distribution of shared (or correlated) versus independent inputs differs when juxtaposing the initial light stimulation phase, which begins to trigger spiking, against subsequent phases. This differentiation might be particularly pertinent to the PV to SC measurements. Here, the initial phase of stimulation, as depicted in Figure 7, reveals a relatively sparse temporal frequency of IPSCs. This might not represent the physiological conditions under which high-firing INs function.While the authors seem to have addressed parts of this concern in their focal stim experiments by examining correlations during both high and low light intensities, they could potentially extract this metric from data acquired in their ramp conditions. This would be especially valuable for PV to SC measurements, given the absence of corresponding focal stimulation experiments.

As the reviewer's comments recognise, the consistent results with focal stimulation already provide direct experimental validation to our ramp stimulation approach. We appreciate the suggestion for further analysis, but as we understand it this analysis would be hard to interpret. First, variation between pairs in the activity at different phases of the light ramp will be confounded by slice to slice differences in the level of ChR2 expression, e.g. in Figure 2, Figure Supplement 1 within slice variability is low, whereas between slice variation is relatively high. This is because in slices with relatively low expression spike onset is relatively late, while in slices with relatively high expression spike onset is early in the ramp and later in the ramp neurons experience depolarising block. Second, the onset of changes in cross-correlation coefficients and lag variation is typically abrupt. This makes it challenging to assign windows to onset phases or to interpret the resulting data.

(3) Re results from Figure 2: Please fully describe the model in the methods section. Generally, I like using a modeling approach to explore the impact of convergent synaptic input to PVs from SCs that could effectively validate the experimental approach and enhance the interpretability of the experimental stim/recording outcomes. However, as currently detailed in the manuscript, the model description is inadequate for assessing the robustness of the simulation outcomes. If the IN model is simply integrate-and-fire with minimal biophysical attributes, then the findings in Fig 2F results shown in Fig 2F might be trivial. Conversely, if the model offers a more biophysically accurate representation (e.g., with conductance-based synaptic inputs, synapses appropriately dispersed across the model IN dendritic tree, and standard PV IN voltage-gated membrane conductances), then the model's results could serve as a meaningful method to both validate and interpret the experiments.

We have expanded the description of the modelling given in the methods including clearer motivation and justification (p 15). Two points are helpful to consider:

First, the goal of the model is to assess the feasibility of the correlation based approach given the synaptic current responses recorded at the soma. We now make this clearer by stating that:

'The goal of our simulations was to assess if analysis of cross-correlations between currents recorded from pairs of neurons could be used to establish whether they receive shared input from the same pre-synaptic neuron. While this should be obvious if neurons exclusively receive shared input, we wanted to establish whether shared input is detectable when each neuron also receives independent inputs of similar frequency and amplitude to the shared input.' (p 15).

The suggestion that the results in Figure 2F are trivial doesn't make sense to us. Indeed, it strikes us as non-trivial that with this approach shared input from a single common presynaptic neuron is not detectable, but input from two or more is.

Second, because we are simulating a somatic voltage-clamp experiment the details of the neuronal time constants, voltage-gated channels or other integrative mechanisms that reviewer suggests may be important here are not actually relevant to the interpretation. To appreciate this consider the membrane equation:CdVmdt+gsyn(t)(Vm−Esyn)+gpass(Vm−Epass)+gVGIC(v,t)(Vm−EVGIC)=0

When the membrane is clamped at a fixed potential, there is no capacitance current (CdVmdt=0), while voltage-dependent ionic currents (gVGIC(v,t)(Vm−EVGIC)) and the resting ionic current (gpass(Vm−Epass)) are constant. In this case the only time varying current is the synaptic current (gsyn(t)(Vm(t)−Esyn)). Thus, adding more details would not make the model more 'meaningful' as these details would be redundant and the results will be the same as simply considering convolution of the synaptic conductances. We have made this rationale clearer in the revised methods (p 15).

**Reviewer #3 (Public Review):**
These are technically demanding experiments, but the authors show quite convincing differences in the correlated response of cell pairs that are close to each other in contrast to an absence of correlation in other cell pairs at a range of relative distances. This supports their main point of demonstrating anatomical clusters of cells receiving shared inhibitory input.

We appreciate the positive comments.

The overall technique is complex and the presentation could be more clear about the techniques and analysis.

Thanks. We've added additional explanation to the methods section to try to improve clarity (p 15-16).

In addition, due to this being a slice preparation they cannot directly relate the inhibitory interactions to the functional properties of grid cells which was possible in the 2-photon in vivo imaging experiment by Heys and Dombeck, 2014.

We agree the two approaches are complementary. The Heys and Dombeck study could only reveal correlations in functional activity, which could have many possible synaptic mechanisms, whereas our results address synaptic organisation but the representational roles of the specific neurons we recorded from are unclear. We have highlighted these current limitations and strategies to address them in the final paragraph of the discussion (p 11).